mSystems

# MbovP0725, a secreted serine/threonine phosphatase, inhibits the host inflammatory response and affects metabolism in *Mycoplasma bovis*

Hui Zhang,[1] Yiqiu Zhang,[2] Doukun Lu,[2] Xi Chen,[2] Yingyu Chen,[2] Changmin Hu,[2] Aizhen Guo[2,3,4,5,6]

**ABSTRACT**  *Mycoplasma* species are able to produce and release secreted proteins, such as toxins, adhesins, and virulence-related enzymes, involved in bacteria adhesion, invasion, and immune evasion between the pathogen and host. Here, we investigated a novel secreted protein, MbovP0725, from *Mycoplasma bovis* encoding a putative haloacid dehalogenase (HAD) hydrolase function of a key serine/threonine phosphatase depending on $Mg^{2+}$ for the dephosphorylation of its substrate *pNPP,* and it was most active at pH 8 to 9 and temperatures around 40°C. A transposon insertion mutant strain of *M. bovis* HB0801 that lacked the protein MbovP0725 induced a stronger inflammatory response but with a partial reduction of adhesion ability. Using transcriptome sequencing and quantitative reverse transcription polymerase chain reaction analysis, we found that the mutant was upregulated by the mRNA expression of genes from the glycolysis pathway, while downregulated by the genes enriched in ABC transporters and acetate kinase–phosphate acetyltransferase pathway. Untargeted metabolomics showed that the disruption of the *Mbov_0725* gene caused the accumulation of 9-hydroxyoctadecadienoic acids and the consumption of cytidine 5′-monophosphate, uridine monophosphate, and adenosine monophosphate. Both the exogenous and endogenous MbvoP0725 protein created by purification and transfection inhibited lipopolysaccharide (LPS)-induced IL-1β, IL-6, and TNF-α mRNA production and could also attenuate the activation of MAPK-associated pathways after LPS treatment. A pull-down assay identified MAPK p38 and ERK as potential substrates for MbovP0725. These findings define metabolism- and virulence-related roles for a HAD family phosphatase and reveal its ability to inhibit the host pro-inflammatory response.

**IMPORTANCE** *Mycoplasma bovis* (*M. bovis*) infection is characterized by chronic pneumonia, otitis, arthritis, and mastitis, among others, and tends to involve the suppression of the immune response via multiple strategies to avoid host cell immune clearance. This study found that MbovP0725, a haloacid dehalogenase (HAD) family phosphatase secreted by *M. bovis*, had the ability to inhibit the host pro-inflammatory response induced by lipopolysaccharide. Transcriptomic and metabolomic analyses were used to identify MbovP0725 as an important phosphatase involved in glycolysis and nucleotide metabolism. The *M. bovis* transposon mutant strain T8.66 lacking MbovP0725 induced a higher inflammatory response and exhibited weaker adhesion to host cells. Additionally, T8.66 attenuated the phosphorylation of MAPK P38 and ERK and interacted with the two targets. These results suggested that MbovP0725 had the virulence- and metabolism-related role of a HAD family phosphatase, performing an anti-inflammatory response during *M. bovis* infection.

**KEYWORDS**  *Mycoplasma bovis*, MbovP0725, HAD phosphatase, transcriptomics and metabolomics, anti-inflammatory response

Address correspondence to Aizhen Guo, aizhen@mail.hzau.edu.cn.

The authors declare no conflict of interest.

See the funding table on p. 21.

*M*ycoplasma species are parasitic bacteria belonging to the Mollicutes class, which are known for their limited genomes and lack of cell walls. It is widely believed that *Mycoplasma* originated from gram-positive bacteria by degenerative evolution. The genus comprises 124 species, 14 of which are human pathogens, while others infect farm animals, herd animals, and pets (1). As such, they are of great importance in both the medical and veterinary fields. In cattle and beef, *Mycoplasma bovis* (*M. bovis*) is an important opportunistic pathogen associated with bovine respiratory disease (BRD) causing a variety of clinical diseases including mastitis, pneumonia, arthritis, otitis media, keratoconjunctivitis, and genital disorders. These diseases have substantial economic implications for the dairy and beef industries, and the use of antibiotic treatment is often ineffective and results in more drug-resistant *M. bovis* isolates, especially to fluoroquinolones, enrofloxacin, and doxycycline antibiotics (2).

As an emerging significant pathogen, *M. bovis* is now spread around the world and has occurred in several countries previously considered free of *M. bovis*, such as Finland, New Zealand, and Argentina, resulting in calf mortality, weight loss in surviving calves, and a decline in milk production in dairy cows (3). In order to decrease the risk of introduction from *M. bovis*-infected cattle, the New Zealand government has culled all *M. bovis*-positive herds (http://www.biosecurity.govt.nz/protection-and-response/mycoplasma-bovis/), and the European Union has funded a DISCONTOOLS project in their disease database (4).

*M. bovis* infection has a long incubation period, and infection is either subclinical or results in mild symptoms. It has traditionally been linked to chronic BRD with characteristic pneumonic lesions and has also exacerbated the acute phase of BRD by increasing the abundance of *Mannheimia haemolytica*, *Pasteurella multocida,* and other viruses (5). The occurrence of these chronic infections suggests that *M. bovis* has strategies to escape host immunity.

The *M. bovis* genome encodes 808 genes, of which 762 encode proteins (6). Through approximately 4,000 random transposon mutagenesis screening analyses of the *M. bovis* genome in a rich medium, 352 genes with essential functions for growth and the remaining non-essential genes acting as putative virulence-related factors have been subsequently identified (7). Due to the lack of a cell wall in *Mycoplasma*, the surface or membrane localization proteins are believed to serve as the primary interface between *M. bovis* and their host. Previous studies have identified some virulence-associated genes of *M. bovis* involved in adhesion (8), invasion (9), peroxide production (10), biofilm formation (11), avoiding phagocytosis of neutrophil granulocytes and macrophages (12), modulating immune responses of the host by immunoglobulin-binding proteins (13), and degrading neutrophil extracellular traps through nucleases (14, 15). Recently, the concepts of *Mycoplasma* secretome and releasome have been proposed, and several secreted proteins were found in many *Mycoplasma* species (16). It is generally assumed that the components released into the culture supernatant or extracellular environment including enzymes, polypeptides, exopolysaccharides, and extracellular vesicles constituted the secretome of *Mycoplasma*. In *Mycoplasma mycoides* subsp. *capri* strain, five secreted proteases were found in the supernatant, and one of them, serine protease S41, was responsible for caseinolytic activity and, after its mutation, resulted in the cleavage of other surface-exposed proteins (17). In *Mycoplasma agalactiae*, a capsular β-(1→6)-glucopyranose is secreted by a glycosyltransferase with synthase activity (18). Previous studies in our laboratory have focused on defining the secretome of *M. bovis* and have identified the biological functions of more secreted proteins during *M. bovis* infection. Sixty secreted proteins from *M. bovis* HB0801 were firstly found using two-dimensional gel electrophoresis and the liquid chromatography-tandem mass spectrometry (LC-MS/MS) proteomic approach (19). Another 178 proteins commonly secreted by *M. bovis* HB0801 virulent and attenuated strains were found using label-free quantitative proteomic analysis (20). Several secreted proteins have been confirmed to have virulence-related activities. For example, MbovP0145 has been shown to induce bovine lung epithelial cells (EBL) to produce IL-8 by interacting with β-actin

(21), MbovP280 induced apoptosis of bovine macrophages (22), and MbovP475 with a conserved transcription activator-like effector decreased cell viability (23). Among these proteins, a differentially expressed protein MbovP0725 attracted attention due to its higher expression in a wild-type (WT) strain than the attenuated strain, indicating that it might play a role in the pathogenesis of *M. bovis*. The *Mbov_0725* gene was annotated as a putative hydrolase of the haloacid dehalogenase (HAD) superfamily, which contains a large number of enzymes including phosphatase, phosphonatase, phosphomutase, and dehalogenase that share a conserved core domain and yet catalyze diverse reactions (24). These proteins are widely distributed in eukaryotic and prokaryotic organisms, and the biochemical roles played within an organism vary from signal transduction to DNA repair to secondary metabolism. There are 28 and 45 genes for the HAD enzymes in the genomes of *Escherichia coli* and the yeast *Saccharomyces cerevisiae*, respectively (24, 25). All HAD superfamily members contain a highly conserved α/β core domain that supports a catalytic scaffold, comprised of four conserved motifs I to IV that position the residues functioning in $Mg^{2+}$ cofactor binding, substrate binding, acid–base, and nucleophilic catalysis (26). The HAD family protein is further divided into three subfamilies, I, II (A and B), and III, according to the topology and insertion of the cap domain that dictates substrate specificity. Subfamily I is characterized by a small α-helical bundle cap domain located between motifs I and II of the core domain. Type II subfamily member cap domains are located between core domain motifs II and III and have two different α/β folds, designated types IIA and IIB. Type III HAD members have only a core domain with a connecting loop serving in place of the cap domain. A previous study reported that several HAD hydrolases were usually conserved in *Mycoplasma* species and participated in metabolite repair to remove phosphorylated damaged or toxic intermediates such as phosphor-sugars (27). In the *M. bovis* HB0801 genome, seven genes belong to the HAD superfamily, and MbovP0725 was also annotated as Cof-type HAD-IIB family protein homology to HAD18 in *E. coli*, but its physiological roles have remained unclear.

This study reports the full biological characterization of the protein encoded by the *Mbov_0725* gene from *M. bovis*, a secreted protein with serine/threonine phosphatase. This study also reports using transcriptomics combined with metabolomics, where *M. bovis* lacking in MbovP0725 (T8.66 strain) showed differential gene expression and metabolite profiles. It shows that MbovP0725 has phosphatase activity that could suppress a pro-inflammatory response induced by lipopolysaccharides (LPSs) and attenuate the activation of MAP kinases in host cells. Immunoprecipitation was used to confirm that MAPK P38 and ERK were its potential interactors.

## RESULTS

### MbovP0725 is a secreted phosphatase of HAD superfamily

The gene Mbov_0725 was predicted to encode a hydrolase of the HAD superfamily. The HAD superfamily members that perform phosphoryl transfer have four signature motifs located in the loops, which carry the conserved catalytic residues. Seventeen amino acid sequences were selected by NCBI after blasting with MbovP0725 (GenBank: AFM52074.1), and it was found that MbovP0725 contained all four of these motifs, as shown in Fig. 1A. Figure 1B shows the expression and purification of recombinant MbovP0725 (rMbovP0275) protein by *E. coli* harboring the plasmid pET-30a. The purified rMbovP0725 protein separated onto SDS-PAGE, stained with Coomassie blue, exhibited a size of 33 kDa and correlated with the predicted size of 32.9 kDa of MbovP0725. It was noted that MbovP0725 could be detected in the supernatant from 12 to 48 h during the growth of *M. bovis in vitro*, as seen in Fig. 1C, with secreted protein MbovP0280 used as a positive control (22).

To determine if the purified rMbovP0725 protein was functional, the phosphatase activity of this protein was assayed using the substrate pNPP. The rMbovP0725 could hydrolyze pNPP in a time-dependent manner, in an alkaline pH of 8.0, as seen in Fig. 1D. Reactions with different doses of rMbovP0725, reactions without this protein, or

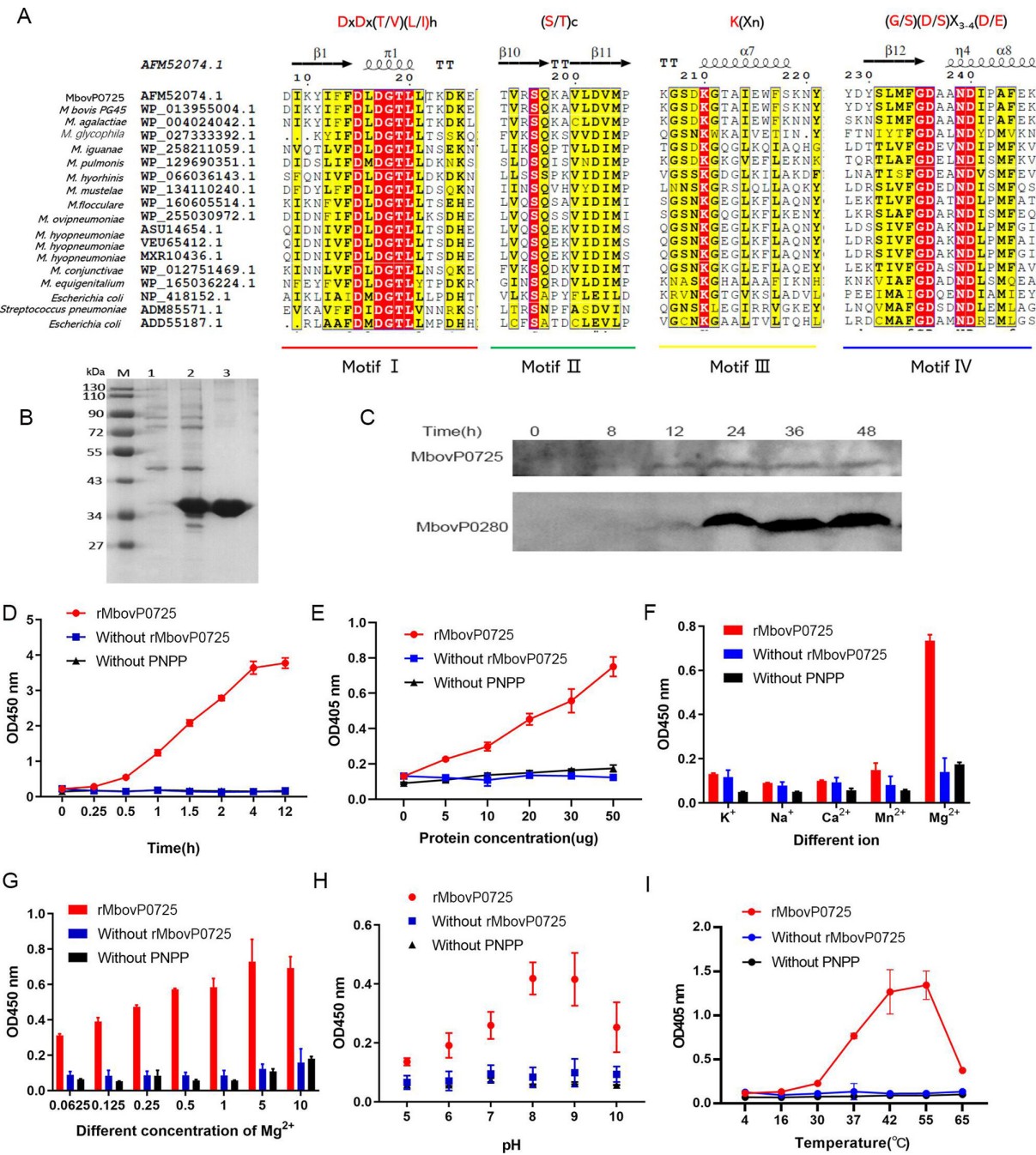

**FIG 1** MbovP0725 is a secreted phosphatase of the HAD superfamily. (A) Proteins homologous to MbovP0725 were identified by hhpred online server and alignment with sequences using CLUSTALW and ESPript. (B) Purification of MbovP0725. The protein was purified with nickel affinity chromatography and resolved with SDS-PAGE. (C) Confirmation of secreted MbovP0725 in culture supernatant. (D) Phosphatase activity of rMbovP0725 (30 μg) with p-nitrophenyl phosphate (pNPP) as substrate in the presence of 5 mM Mg$^{2+}$ for 0, 0.25, 0.5, 1, 1.5, 2, 4, and 12 h. (E) Phosphatase activity of different concentrations of rMbovP0725. Various amounts (0, 5, 10, 20, 30, and 50 μg) of rMbovP0725 protein were added to the reaction buffer containing pNPP and 5 mM Mg$^{2+}$. (F) Effect of different monovalent and divalent cations (K$^+$, Na$^+$, Ca$^{2+}$, Mn$^{2+}$, and Mg$^{2+}$) on the phosphatase activity of rMbovP0725. (G) Effect of different concentrations of Mg$^{2+}$ (0.0625, 0.125, 0.25, 0.5, 1, 5, and 10 mM) on the phosphatase activity of rMbovP0725. (H) Effect of different pHs (5, 6, 7, 8, 9, and 10) on the phosphatase activity of rMbovP0725. (I) Effect of different temperatures (4, 16, 30, 37, 42, 55, and 65°C) on the phosphatase activity of rMbovP0725. Without rMbovP0725 or pNPP was used as a negative control (NC). Values represent the mean ± SD.

reactions without pNPP in the reaction mixture showed no color, indicating that the protein was dose-dependent, as shown in Fig. 1E. It also showed strict dependency on metallic Mg$^{2+}$ seen in Fig. 1F, where the optimal concentration of Mg$^{2+}$ in the reaction

system was approximately 5 mM (Fig. 1G). The highest reaction rates were observed between pH 8.0 and 9.0, as shown in Fig. 1H. The rMbovP0725 was a relatively thermostable phosphatase, with an optimal reaction temperature of about 55°C, as seen in Fig. 1I. Combined, MbovP0725 is a secreted phosphatase of the HAD superfamily.

## MbovP0725 is an active serine/threonine phosphatase

To determine the substrate specificity for purified His$_6$-tag fusion protein rMbovP0725, three different types of molecules including O-phosphoserine (p-Ser), O-phosphothreonine (p-Thr), and O-phosphotyrosine (p-Tyr) that were identified as substrates for rMbovP0725 were tested using the malachite green assay (Abnova, USA). Of these, p-Ser and p-Thr were the optimal substrates for rMovP025, suggesting that rMbovP0725 was a specialized serine/threonine phosphatase, as shown in Fig. 2A. To confirm the involvement of the predicted active site of MbovP0725 catalytic activity, site-directed mutagenesis was performed, and the resultant MbovP0725 mutants were tested *in vitro* for activity toward *p*NPP substrates as described above. In accordance with their predicted contributions to catalysis, the D15A, T48A, K210A, and D236A mutations prevented phosphatase activity, validating their importance for the catalytic function of this protein.

## MbovP0725 could decrease adhesion and pro-inflammatory response induced by LPS during *M. bovis* infection

MbovP0725 was deleted from *M. bovis* HB0801 (T8.66) based on the transposon *M. bovis* mutant library previously prepared in this laboratory (28) and correspondingly complemented (CT8.66) in this study. Western blotting assays with the wild-type (WT) HB0801, T8.66, and CT8.66 strains and the recombinant membrane protein MbovP579 used as the control confirmed that MbovP725 expression was deficient in T8.66 but present in both WT HB0801 and CT8.66, as shown in Fig. 3A. The deletion of Mbov_0725 gene did not affect the growth and morphology phenotype of the three *M. bovis* strains in the pleuropneumonia-like organism (PPLO) medium *in vitro,* as seen in Fig. 3B and C.

The BoMac cells were infected with HB0801, T8.66, and CT8.66 at a multiplicity of infection (MOI) of 1,000 for 12 h, and the cytokines were assessed with qRT-PCR assays. Compared with wild-type and complement strain CT8.66, T8.66 significantly increased the expression of IL-1β, IL-6, and TNF-α, shown in Fig. 3D-F. The results showed that MbovP0725 could inhibit the pro-inflammatory response during *M. bovis* infection of the host, and its adhesion ability was reduced in EBL and MAC-T epithelial cells, as seen in Fig. 3G, H.

## MbovP0725 disruption in *M. bovis* induces gene expression changes involved in glycolysis pathway, mismatch repair system, and acetate kinase–phosphate acetyltransferase pathway

To analyze the effect of MbovP0725 on *M. bovis* gene expression, a transcriptome analysis was performed. The transcriptome profiles of three biological replicates for both groups were separated on the principal component analysis (PCA) graph, where PC1 and PC2 showed 85.31% and 4.9% variation, respectively. Compared with the wild-type strain HB0801, based on the criteria of $P < 0.05$, fold change >1, 228 differentially expressed genes (DEGs) were marked as upregulated and 226 DEGs were downregulated in the T8.66 mutant strains, as seen in Fig. 4A. Gene ontology (GO) enrichment classification of differentially expressed genes was also performed, as shown in Fig. 4B. Three functional aspects were included in the GO analysis, including biological process, molecular function, and cellular component. Compared with HB0801, the small molecule metabolic processes in biological processes and ion binding in molecular function were significantly affected by MbovP0725.

KEGG analysis revealed the significantly enriched pathway as presented in Fig. 4C. Among the most enriched pathways, the pathway involved in genetic information

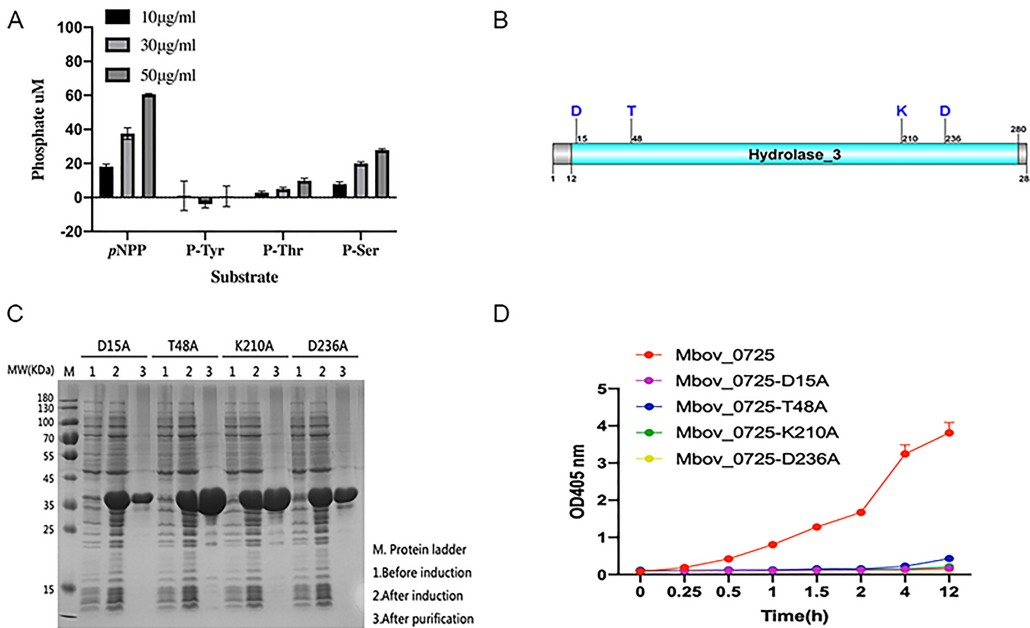

**FIG 2** MbovP0725 is a serine/threonine phosphatase. (A) Comparison of the substrate preferences of MbovP0725. Amounts of released inorganic phosphate were measured by the malachite green assay. The mean values of the three replicates are shown. (B) Schematic representation of the predicted key phosphorylation sites of MbovP0725. (C) Bacterial expression and purification of the site-directed mutagenesis for D15A, T48A, K210A, and D236A of MbovP0725. (D) Phosphatase activity of MbovP0725 and its mutant protein with pNPP as substrate. Values represent the mean ± SD.

processing was significantly impacted as well as microbial metabolism in diverse environments, pyruvate and pyrimidine metabolism, carbon metabolism, glycolysis, and biosynthesis of secondary metabolites. The deletion of Mbov_0725 increased 12 genes involved in glycolysis and mismatch repair system in compensation and decreased three genes including *Mbov_0567*, *Mbov_0568,* and *Mbov_0273* in the acetate kinase (Ack)– phosphate acetyltransferase (Pta) pathway, as shown in Fig. 4D. Interestingly, we found that two genes, *Mbov_0799* and *Mbov_0800,* from the type IV TA system AbiEii/AbiGii toxin and AbiEi antitoxin families were not expressed in the Mbov_0725 mutant strain T8.66 according to the transcriptome data and also confirmed using qRT-PCR (their CT values were too high to be determined).

## Metabolomic profiling revealed significant differences between WT HB0801 and MbovP0725 mutant in a number of metabolites of nucleotide metabolism

To investigate whether and how MbovP0725 altered *M. bovis* metabolism, untargeted metabolomic analysis was performed, where the MbovP0725 mutant and *M. bovis* strain were grown in a PPLO medium and harvested at the same point of log-phase growth. A total of 8,545 and 2,112 ion peaks under positive and negative modes were collected, respectively. A total of 10,657 compounds were detected in the samples, of which 2,678 were annotated as known metabolites. To analyze differentially accumulated metabolites (DAMs) between groups, PCA and partial least-squares discriminant analysis (PLS-DA) were used. The PCA score plot showed a clear distinction between the HB0801 strain and the MbovP0725 mutant in both positive and negative modes, as shown in Fig. 5A and B. A total of 3,593 differential metabolites with significant differences were identified in the HB0801 and MbovP0725 mutant strains in both positive and negative modes, among which 1,459 were upregulated (fold changes ≥1.2) and 2,134 were downregulated (fold change ≤0.8). The absence of MbovP0725 greatly affected *M. bovis* metabolites associated with cysteine and methionine metabolism, biosynthesis of amino acids,

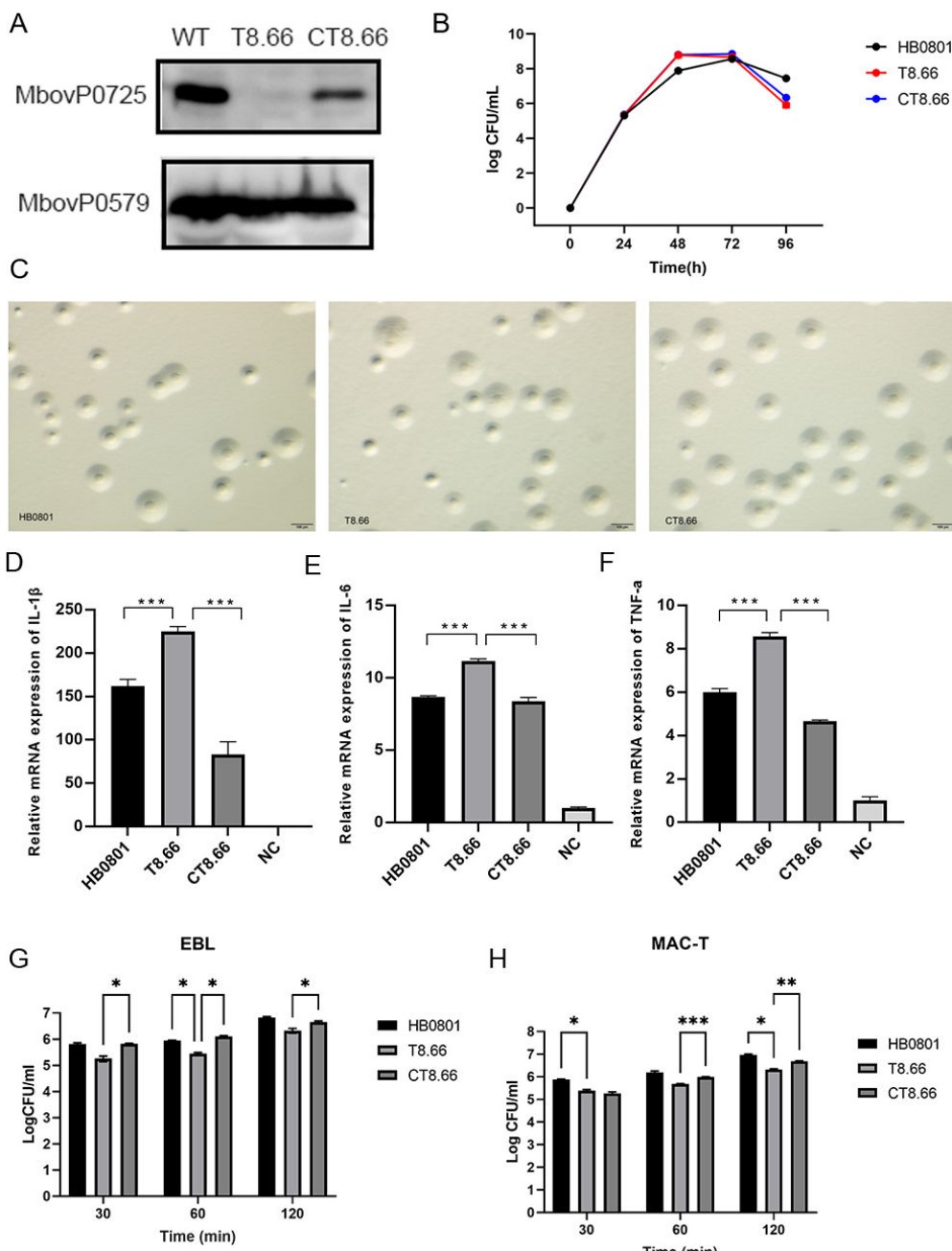

**FIG 3** MbovP0725 mutant strain T8.66 elicited a stronger inflammatory response of BoMac cells and less adhesion to epithelial cells compared to the wild-type strain and its complement strain CT8.66. (A) Visualization of MbovP0725 expression in HB0801, T8.66, and CT8.66 using western blotting assay. (B) Growth curves of HB0801, T8.66, and CT8.66 strains. Growth of *M. bovis* at each time point was determined with a CFU plating assay. (C) Microscopic view of the colony morphology of HB0801, T8.66, and CT8.66 strains. (D–F) MbovP0725 disruption could enhance the mRNA expression of pro-inflammatory cytokines (IL-1β, IL-6, and TNF-α) in response to BoMac infected with different strains, assessed by quantitative reverse transcription polymerase chain reaction (qRT-PCR). (G, H) T8.66 decreased its adhesion ability compared to WT strain and CT8.66. The binding to EBL and MAC-T cells was determined by counting the number of CFU associated with cell monolayers following incubation with $10^8$ CFU. The data are the mean *Mycoplasma* titers from three independent assays. Two-way ANOVA was used to determine the statistical significance of differences between the treatments. $^*P < 0.05$, $^{**}P < 0.01$, and $^{***}P < 0.001$ indicate statistically significant differences.

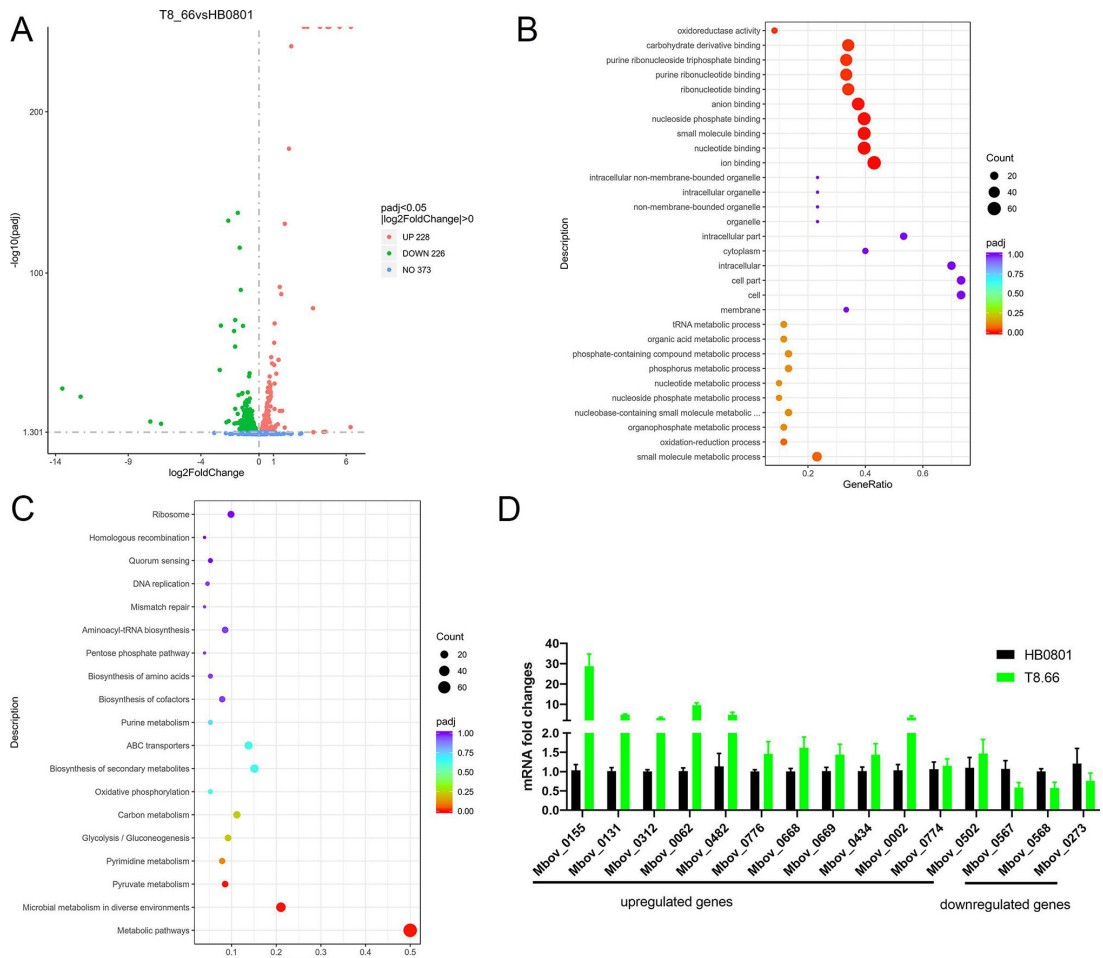

**FIG 4** Transcriptome analysis of the DEGs in T8.66 compared with WT HB0801. (A) The volcano map of DEGs between T8.66 and HB0801. The *x*-axis is the log2 fold change, and the *y*-axis is the −log10 (*P*-value). The horizontal line is the threshold of *P*-value =0.05. Red dots indicate gene upregulation, and green dots indicate gene downregulation. Gray dots denote genes with no significant change. (B) Bubble chart of GO enrichment analysis of DEGs. *P*-values range from 0 to 1. The size of the dot denotes the number of DEGs. The gene ratio is the number of the DEGs to the number of annotated genes. (C) Bubble chart of Kyoto Encyclopedia of Genes and Genomes (KEGG) pathway analysis. The *y*-axis is the different KEGG pathways, and the *x*-axis is the gene ratio, which represents the ratio of the number of DEGs to the total number of annotated genes in this pathway. The size of the dot correlates with the number of DEGs annotated in the pathway. (D) Validation of RNA-seq results by qRT-PCR.

microbial metabolism in diverse environments, purine metabolism, thiamine metabolism, and ABC transporters, as seen in Fig. 5C and D. As shown in the volcano map in Fig. 5E and F, DAMs were annotated to KEGG pathways, and the KEGG analysis showed that these were mainly enriched in 15 pathways. The significantly impacted pathways included pyrimidine and purine metabolism, biosynthesis of amino acids, cysteine and methionine metabolism, microbial metabolism in diverse environment, degradation of aromatic compounds, carbon metabolism, glyoxylate and dicarboxylate metabolism, and biosynthesis of secondary metabolites. Most of these pathways are related to nucleotide and amino acid metabolisms.

Among the 20 upregulated metabolites with high VIP scores in both positive and negative modes, T8.66 showed significantly higher abundances of 9-HODE and 4-phenylbutyric acid, as seen in Fig. 6. The former is oxidized free fatty acids, one of the major lipid components of oxidized low-density lipoproteins, and the latter is a monocarboxylic acid, which inhibits protein isoprenylation. When MbovP0275 disruption in T8.66 occurred, the intermediate products in purine and pyrimidine metabolic

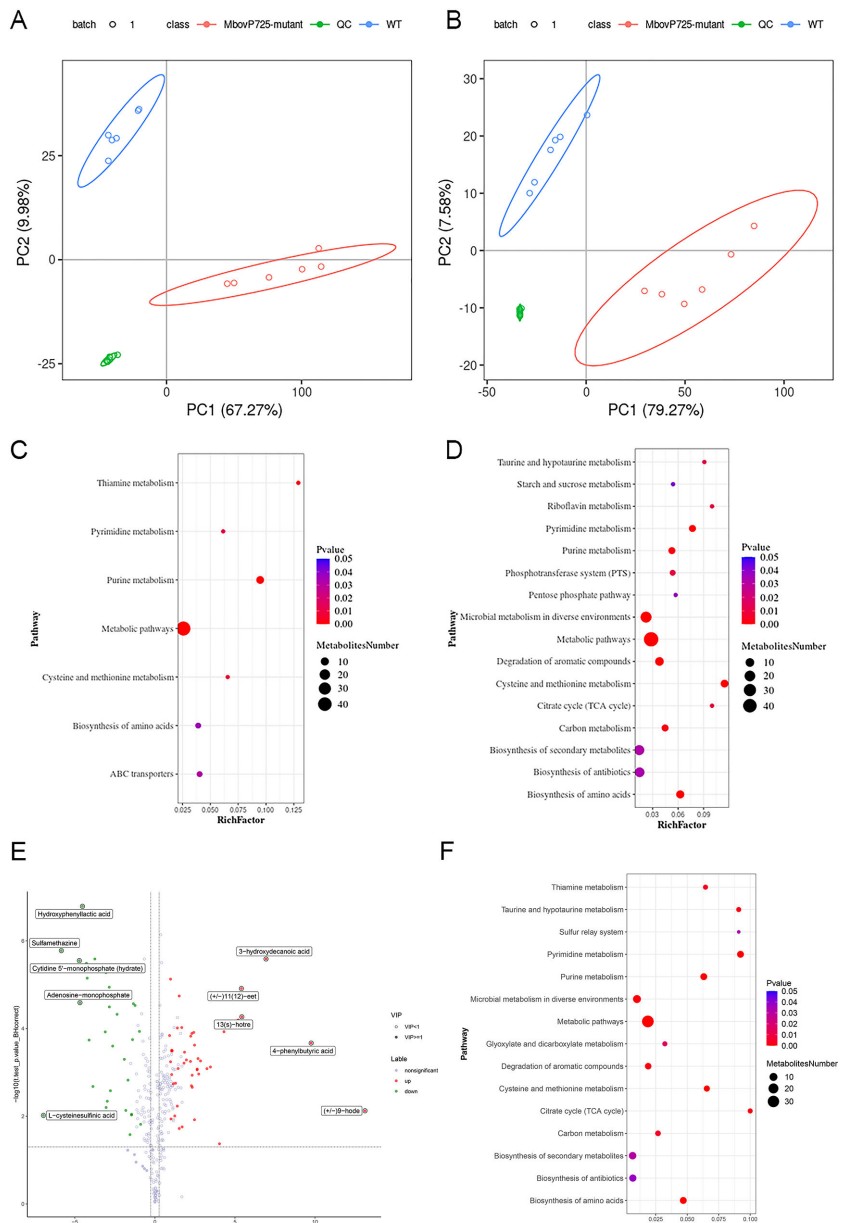

**FIG 5** The metabolomic analysis of MbovP0725 activity in *M. bovis*. (A and B) PCA score chart for all samples in pos (A) and neg (B) modes. The abscissa is PC1, and the PCA score graph shows 95% confidence intervals. Each dot represents a sample, and different groups are labeled with different colors. Six replicates were included for both HB0801 and T8.66. (C and D) Bubble plots for metabolic pathway enrichment analysis in positive (C) and negative (D) modes. The *x*-axis enrichment factor (RichFactor) is the number of differential metabolites annotated to the pathway divided by all identified metabolites annotated to the pathway. The larger the value, the greater the proportion of differential metabolites annotated to the pathway. The dot size represents the number of DAMs annotated to this pathway. (E) Volcano map of DAMs in both positive and negative modes. The log2 of fold change plotted against the −log10 of the *P*-value. The two vertical dotted lines in the figure are the two fold change threshold, and the horizontal dotted line is the *P*-value at a 0.05 threshold. Red and green dots represent up- and downregulated DAMs, respectively, and gray dots represent non-significant metabolites. A variable with a VIP score close to or greater than 1 was considered important in the model. VIP, variable importance in projection. (F) Bubble chart of KEGG pathways involving significantly enriched DAMs in both pos and neg modes.

pathways such as adenosine 5′-monophosphate (AMP), uridine monophosphate (UMP), and cytidine 5′-monophosphate (CMP) were downregulated, as also shown in Fig. 6.

## MbovP0725 inhibited pro-inflammatory cytokine production

To reveal the inflammatory response of recombinant MbovP0725 protein on the host cell, IL-1β, IL-6, and TNF-α mRNA expression induced by LPS- and protein-treated models were measured in BoMac and MAC-T cells, respectively. The results showed that after induction by LPS, exogenous rMbovP0725 protein treatment could significantly decrease the production of IL-1β, IL-6, and TNF-α with the concentration from 1 to 20 µg/mL after 12 h in the BoMac and MAC-T cells compared to LPS treatment alone, as shown in Fig. 7A and B.

Intracellular MbovP0725 protein was obtained by the transfection of an MbovP0725-recombinant eukaryotic vector pEGFP-N1-MbovP0725 into BoMac or MAC-T cells, respectively. As shown in Fig. 8A and B, intracellular MbovP0725 expression also decreased pro-inflammatory cytokine mRNA expression including IL-1β, IL-6, and TNF-α, at mRNA levels with transfection of the recombinant vector compared to the empty vector after LPS treatment for 12 h. Combined with the result of extracellular MbovP0725 protein treatment, these results suggested that MbovP0725 protein exhibited anti-inflammatory activity.

## The anti-inflammatory function of MbovP0725 is related to its phosphatase activity

To investigate whether the anti-inflammatory function of MbovP0725 depends on its phosphatase activity, MAC-T cells were treated with the endotoxin-removal wild-type rMbovP0725 and its catalytically inactive mutant protein rMbovP0725-D15A upon the stimulation by LPS for 12 h; the mRNA expression level of IL-1β, IL-6, and TNF-α was detected by qRT-PCR. As shown in Fig. 9, wild-type MbovP0725 significantly inhibited IL-1β, IL-6, and TNF-α expression compared with the blank and D15A group after LPS treatment. When combined, the results identified that the anti-inflammatory function of rMbovP0725 was related to its phosphatase activity.

## MbovP0725 attenuates activation of MAPK pathway

Having determined that MbovP0725 was able to decrease IL-1β, IL-6, and TNF-α mRNA expression of host cells, a possible signal pathway modulated by MbovP0725 was further detected, where MAC-T cells were treated with endotoxin-removal rMbovP0725 or MbovP0725 D15A mutant protein, and the changes in the phosphorylation state of P38 and ERK1/2 were tested after stimulation with LPS for 2 h. Western blot analysis for total P38, ERK1/2, phospho-P38, and phospho-ERK1/2 in Fig. 10A, with their gray values shown in Fig. 10C and D, showed that rMbovP0725 significantly attenuated the phosphorylated expression of P38 and ERK1/2 in a dose-dependent manner. However, MbovP0725 with D15A mutation lost the ability to suppress the phosphorylation of P38 and ERK proteins (Fig. 10B, E, and F). To further investigate the potential interaction between MbovP0725 and MAPK, HEK-293T cells were transfected with pCAGGS-HA-MbovP0725, its point mutant protein pCAGGS-HA-MbovP0725-D15A, and HA-empty vector, respectively, followed by the immunoprecipitation (IP) assay. The results showed that only HA-MbovP0725 could be detected in the immunocomplexes isolated with anti-P38 and anti-ERK antibody, as shown in Fig. 10G.

## DISCUSSION

We previously identified and explored the *M. bovis* secretome using software prediction and different proteomics methods, including two-dimensional gel electrophoresis and label-free quantitative proteomics (19, 20). However, until now, the molecular function of these potential secreted proteins has rarely been investigated. Inflammatory diseases

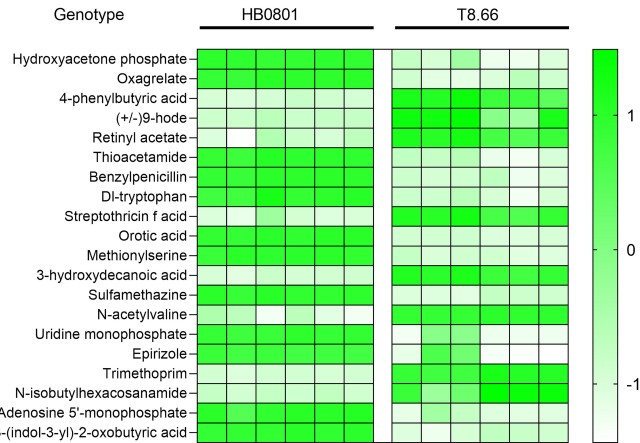

**FIG 6** Heatmap including 20 metabolites from MbovP0725 mutant metabolomic analysis with higher VIP scores. Samples and genotype are represented in columns. High-intensity measurements compared to average intensity are green, and low-intensity measurements are represented by white.

such as pneumonia, mastitis, and arthritis due to *M. bovis* infection represent important issues worldwide, which leads to significant economic losses and affects animal welfare.

## A. BoMac

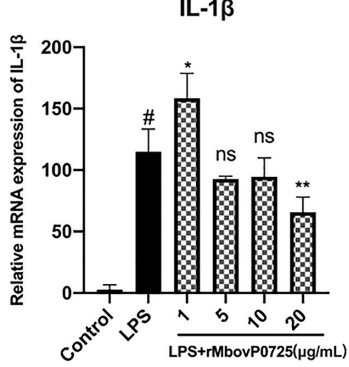
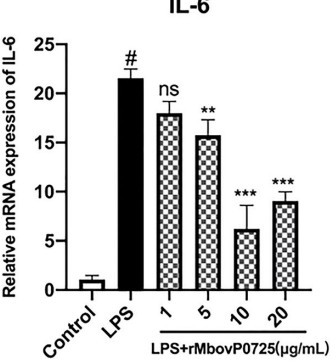
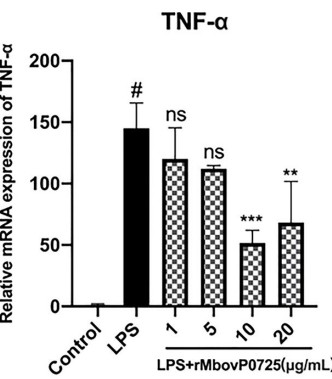

## B. MAC-T

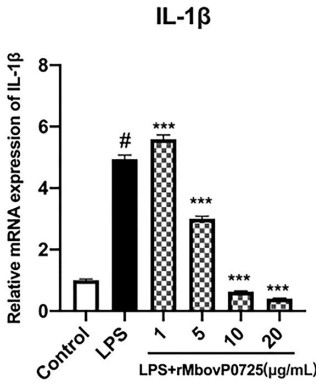
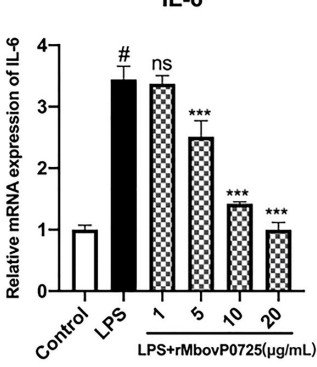
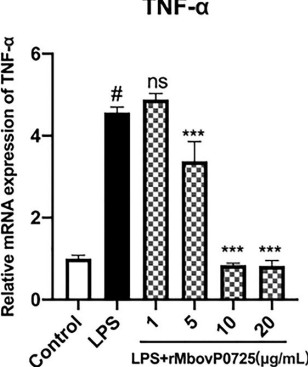

**FIG 7** MbovP0725 inhibited pro-inflammatory cytokine production. (A and B) IL-1β, IL-6, and TNF-α mRNA levels of BoMac and MAC-T cells treated with rMbovP0725 protein at 1, 4, 10, and 20 µg/mL and LPS at 1 µg/mL were measured by qRT-PCR. Two-way ANOVA was used to determine the statistical significance of differences between the treatments. Samples without LPS or MbovP0725 treatment were used as controls. $^{*}P < 0.05$, $^{**}P < 0.01$, and $^{***}P < 0.001$ indicate statistically significant differences.

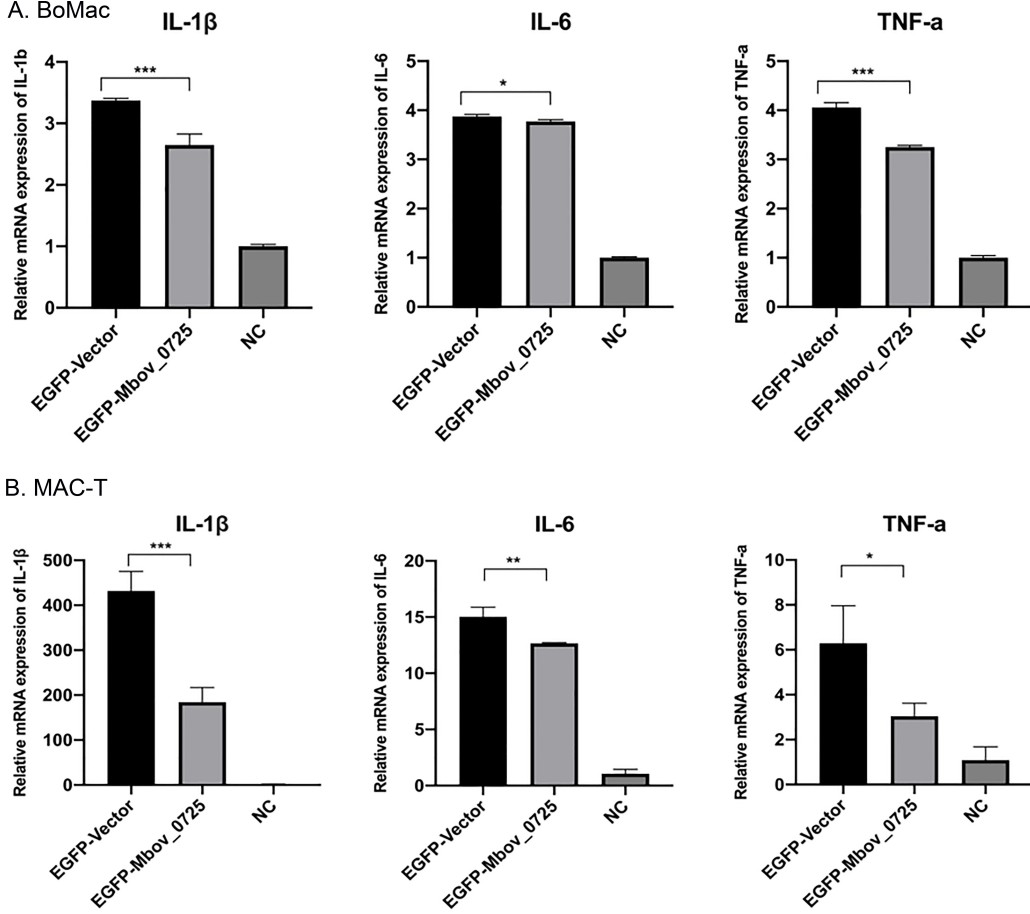

**FIG 8** Intracellular MbovP0725 expression inhibited pro-inflammatory cytokine gene production. Recombinant eukaryotic vector pEGFP-MbovP0725 transfection decreased cytokine mRNA expression including IL-1α, IL-6, and TNF-α in (A) BoMac and (B) MAC-T cells. EGFP-vector mean cells transfected with empty vector. NC indicates PBS-treated cells. $^*P < 0.05$, $^{**}P < 0.01$, and $^{***}P < 0.001$ indicate statistically significant differences.

However, the mechanisms by which *M. bovis* persists and maintains chronic infections in cattle and beef are largely unknown. In terms of the pathogenic mechanism, pathogens such as *M. bovis* can suppress the inflammatory response in the host to ensure successful

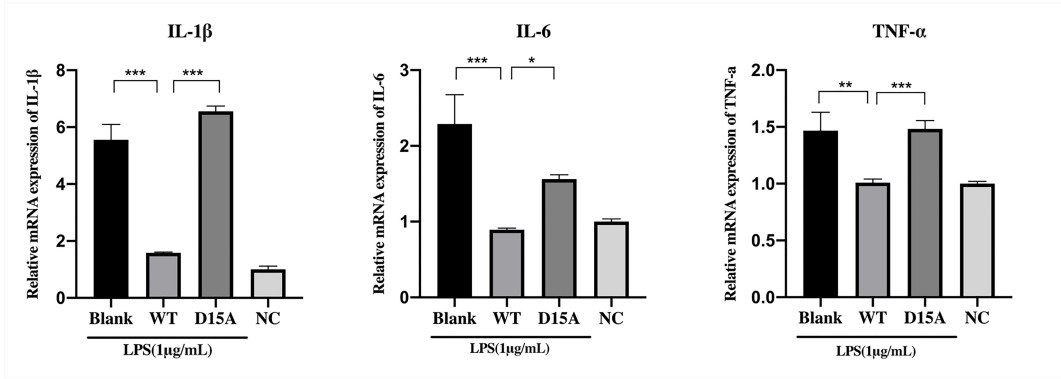

**FIG 9** MbovP0725 suppresses pro-inflammatory response depending on its phosphatase activity. Wild-type rMbovP0725 protein was able to reduce the mRNA expression response to 1 μg/mL LPS treatment, whereas its mutant D15A constructs exhibited reduced ability to suppress the inflammatory response similar to the blank group with LPS treatment alone. NC means no LPS stimulation. Data are the means of three independent assays. Standard deviations are indicated by error bars. In the figure, *P*-values are indicated by asterisks: $^*P < 0.05$, $^{**}P < 0.01$, and $^{***}P < 0.001$.

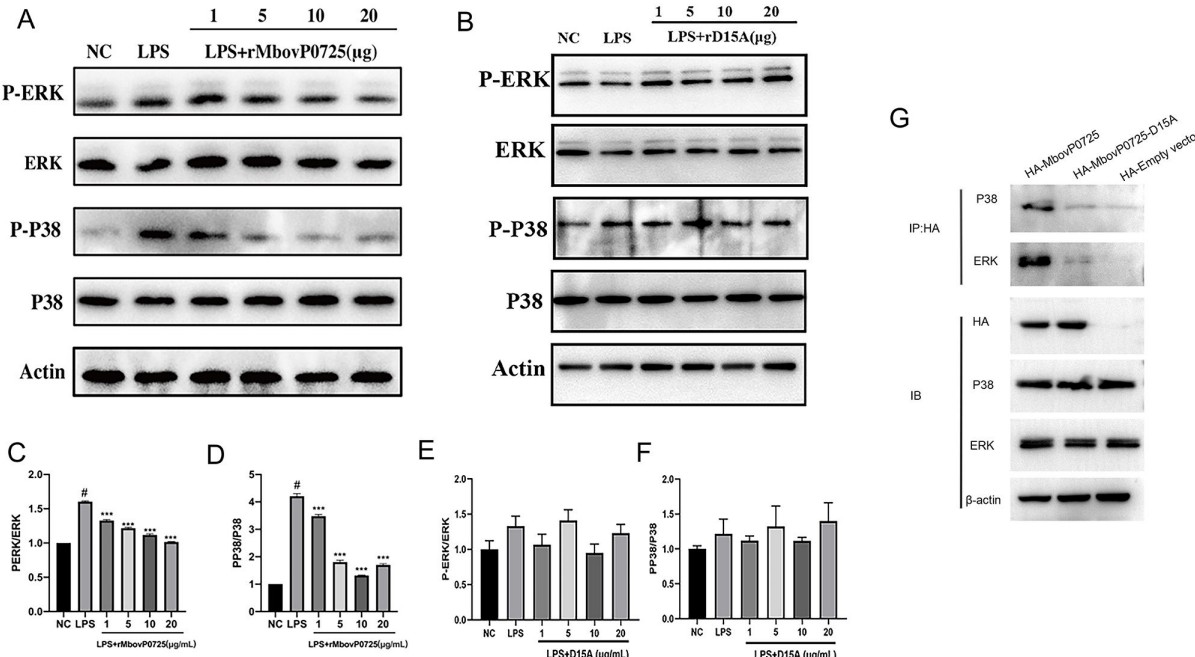

**FIG 10** MbovP0725, but not its mutant protein D15A, attenuated the activation of eukaryotic MAPK pathways induced by LPS. (A and B) Western blot analysis of activation and phosphorylation of MAPK P38 and ERK1/2 was achieved by incubating MAC-T cells with different doses of rMbovP0725 (A) or its mutant protein D15A (B) after stimulation with LPS for 2 h. (C and D) Densitometry qualification of phosphorylated P38 and ERK1/2 was normalized by total P38 and ERK ($^{\#}P <$ 0.001 vs NC group; $^{***}P <$ 0.001 vs LPS group). (E and F) The gray value of phosphorylated P38 and ERK1/2 levels was conducted by normalizing to total P38 and ERK. (G) Pull-down assays showed P38 and ERK as interactors of MbovP0725. The 293T cells transfected with pCAGGS-HA-MbovP0725, its point mutant protein D15A, and HA-empty vector, respectively, following the whole cell lysates were immunoprecipitated with anti-HA tagged antibody and immunoblotted with P38 and ERK antibody. IB, immunoblotting.

infection although the process is complex (29). In this study, we revealed a novel secreted protein, MbovP0725 from *M. bovis*, which demonstrates phosphatase activity specificity for serine and threonine and inhibits inflammatory immunity by attenuating MAPK phosphorylation. MbovP0725 can also affect *M. bovis* glycolysis, *Pta/AckA* pathway, and nucleotide and amino acid metabolisms, as determined by transcriptomics and metabolomics. Interestingly, the phosphatase activity of MbovP0725 not only was related to its anti-inflammatory actions but also influenced its interaction with MAPK P38 and ERK. Ultimately, we describe a bacteria HAD family phosphatase that is important for metabolism of *M. bovis* and inflammatory response of host cells during *M. bovis* infection.

## Feature of MbovP0725 as HAD-type phosphatases in *M. bovis*

The HAD superfamily is primarily composed of phosphatases, which are known to affect a wide range of substrates, including sugars, small metabolites, and proteins, with their specificity often being determined by the "cap" domain (30–32). In this study, MbovP0725 was annotated as a Cof-type HAD-IIB family protein, whose cap domains are located between core domain motifs II and III according to sequence analysis. We found that MbovP0725 demonstrates strong specificity for substrates such as *p*-Ser and *p*-Thr. Serine and threonine are the predominant phosphorylation sites in *Mycoplasma* and other bacteria (33, 34). Phosphorylation and dephosphorylation mediated by kinase and phosphatase are critical to signal transduction pathways. Although *Mycoplasma* has limited protein coding capacity, it has been shown to have post-translational protein modifications, mainly phosphorylation modifications (35). The HAD superfamily is characterized by four conserved motifs in HAD enzymes and acts as phosphatases, phosphomutase, ATPases, and phosphatases, although this enzyme family derives their name from different bacterial hydrolytic dehalogenases (36). Functional studies have

shown that *M. bovis* MbovP0725 is a secreted serine/threonine phosphatase of the HAD superfamily that required $Mg^{2+}$ to interact with pNPP. In this study, we predicted with bioinformatical assay phosphorylation sites within catalytic residues at 15, 48, 210, and 236 in MbovP0725. Subsequently, we performed alanine substitution at 15, 48, 210, and 236 sites in MbovP0725, and two alanine substitutions at residues 48 and 210 significantly decreased, while another two alanine substitutions at 15 and 236 abolish its phosphatase activity indicating that 15 and 236 residues mostly play an important role in regulating its phosphorylation process. During the process of expression and purification, we observed partial degradation of the D236A mutant suggesting that this mutation affected protein stability. Therefore, we finally selected the D15A mutant protein for further study.

Referring to other bacteria, HAD family protein could mediate metabolic and pathogenic interactions with host cells. As in *Legionella pneumophila*, Lem4 has been reported as a tyrosine phosphatase belonging to the HAD family that shows strong specificity for phenylphosphate, as well as an effector involved in interactions with *Legionella*-containing vacuoles or plasma membrane phosphorylated host targets (37). In *Porphyromonas gingivalis*, a secreted phosphoserine phosphatase SerB653 belonging to the HAD family was found to be involved in bacteria internalization and persistence in contact with gingival epithelial cells (38). Moreover, the *S. cerevisiae* Pho13p has been identified as a HAD family phosphatase that could preferentially phosphorylate carbohydrates preferentially on 2-phosphoglycolate (39). In *Mycoplasma flocculare*, dephospho-coenzyme A kinase DPCK, containing a HAD-like domain as cofactor phosphatase, has been shown to affect the activity of the Cof protein by acting on intermediates in thiamine biosynthesis (40). To further explore the function of MbovP0725, we constructed a MbovP0725 mutant strain, T8.66, and its complement strain, CT8.66. Compared to the wild-type strain, the mutant strain elicited a stronger cellular response from macrophages, together with increased expression of pro-inflammatory cytokines (IL-1β, IL-6, and TNF-α), but less adhesion to epithelial cells. In terms of the reasons for these findings, we speculate that MbovP0725, as a phosphatase, alters the metabolism and pathogenicity of *M. bovis*. Therefore, we performed transcriptomic and metabolomic analyses to identify the differential genes and metabolites from *M. bovis* WT and T8.66 strains. Our results revealed the precisely targeted functional category of DEGs responsible for glycolysis and the *Pta/AckA* pathways. After Mbvo_0725 deletion, the expression of the key glycolytic genes, including *pgk*, *pgm*, *eno*, *pk*, *ldh*, and *pdhA/B*, was upregulated, while that of the *pta* and *ackA* genes was downregulated. As reported, many glycolytic enzymes in *Mycoplasma* are localized on the cell surface and act as extracellular matrix-binding proteins involved in cytoadherence and modulating the host immune system. *Mycoplasma gallisepticum Pdha/Pdhb* are reported to be adhesions and surface-exposed immunogenic proteins (41). Eight glycolytic enzymes in *Mycoplasma pneumoniae* including *Pdh-A,B,C, GapA, Ldh, Pgm, Pyk,* and *Tkt* were confirmed as being surface-expressed and able to interact with plasminogen associated with adhesion and invasion (42). Additionally, most of these proteins were metabolic enzymes involved in carbohydrate and energy metabolism. These glycolytic genes have been previously reported to be phosphorylated in *M. pneumoniae* for ATP generation (33). Moreover, the *Pta-AckA* pathway plays an important role in bacterial fitness through the control of central metabolism to sustain balanced growth and cellular homeostasis (43). In *Staphylococcus aureus*, the inactivation of the *Pta–AckA* pathway has been shown to increase the rate of glucose consumption, leading to a metabolic block at the pyruvate node and enhanced carbon flux through glycolysis. Therefore, we speculate that the upregulated genes in glycolytic pathways and the downregulated genes in pta/AckA pathways increase in a compensatory manner to supply sufficient energy for the intracellular survival of *M. bovis*, which may induce stronger anti-inflammatory effects contributing to immune escape. These observations showed considerable agreement with literature reports on HAD enzyme phosphoglycolate phosphatase in *Plasmodium*

*falciparum* mediating metabolite repair that maintains the intermediate products in the glycolytic pathway (44).

The metabolomic data for the MbovP0725 mutant were analyzed in detail by comparing the mutant T8.66 to the WT HB0801 by untargeted metabolomics; 20 differential metabolites with high VIP scores were enriched in nucleotide metabolism, amino acid metabolism, and carbon metabolism, among others. The MbovP0725 mutant strain T8.66 showed a significant depletion of nucleotides for nucleotide biosynthesis including AMP, GMP, UMP, and CMP. Meanwhile, several genes that participated in purine and pyrimidine metabolism, including *Mbov_0206*, *Mbov_0668, cpdB, deoD, Mbov_0051,* and *Mbov_0255*, were also upregulated, suggesting that MbovP0725 affects nucleotide metabolism. These observations showed considerable agreement with literature reports on phosphatase activity against nucleotides in *Mycoplasma mycoides capri* and JCVI-Syn3.0 strain, suggesting that HAD enzymes have nucleotide phosphatase activity (45). In the minimized *Mycoplasma* JCVI-Syn3A genome, four HAD genes encoding stand-alone members of the HAD hydrolase family, including the JCVISYN3A_0728 enzyme, hydrolyzed a wide range of nucleoside and sugar phosphates, the JCVISYN3A_0907 and JCVISYN3A_0077 enzymes hydrolyzed narrower ranges of sugar phosphates, and the JCVISYN3A_0066 enzyme hydrolyzed flavin mononucleotide and CoA (45). Moreover, we observed that the two metabolites, 9-HODE and 4-phenylbutyric acid, with the most differentiation were accumulated in the MbovP0725 mutant strain. The former is oxidized free fatty acids, one of the major lipid components of oxidized low-density lipoproteins, while the latter is a derivative of butyric acid produced by bacterial fermentation. A previous study demonstrated that 9-HODE is a pro-inflammatory mediator through the GPR132 receptor on monocytes and macrophages (46). In addition, 4-phenylbutyric acid is known to inhibit ER stress and reduce the levels of inflammatory mediators (47). Interestingly, some of the glycolytic enzymes have multiple functions in metabolism. For example, the glycolytic kinase is also active in nucleotide phosphorylation, thus replacing the nucleoside diphosphate kinase (48). *M. pneumoniae* possesses a lactate dehydrogenase; alternatively, the NAD oxidase may directly oxidize NADH2 for the regeneration of NAD (49). Although both differential genes and metabolites are involved in the glycolytic pathway and ATP generation, the function of their metabolic pathways and inflammatory response in *M. bovis* remain undefined.

## The anti-inflammatory response of MbovP0725 via attenuated MAPK phosphorylation during *M. bovis* infection

Serine/threonine protein phosphatases are usually divided into three major families as phosphoprotein phosphatases, metal-dependent protein phosphatase, and aspartate-based phosphatases represented by TFIIF-associating component of RNA polymerase II CTD phosphatase/small CTD phosphatase (FCP/SCP) and HAD enzymes (50). Among them, HAD family proteins contain the signature sequence motif DxDxT and have a similar active site with FCP/SCP. In recent years, phosphorylation of serine, threonine, and tyrosine residues has also been shown to be widely distributed in prokaryotes, especially in pathogenic bacteria, and several phosphatases were reported including *Mycoplasma* like prpC in *Mycoplasma synoviae* (51), MG_207 in *Mycoplasma genitalium* (52), and prpC and prkC in *M. pneumoniae* function, correlating with bacteria gliding motility (53).

Furthermore, we found that exogenous MbovP0725 and intracellular MbovP0725 expression reduced the mRNA level of IL-1β, IL-6, and TNF-α and decreased the expression of phosphorylated p38 and ERK1/2 MAP kinase. However, it is currently difficult to detect pro-inflammatory cytokines in BoMac via Enzyme linked immunosorbent assay (ELISA) and western blotting assay due to their low release levels. Additionally, there are a few sensitive commercial ELISA kits for bovine cytokine detection. Therefore, we speculate less IL-1β, IL-6, and TNF-α secreted probably because of the limited current bovine cell lines and insensitive ELISA kits. Further study will be needed to prove the phenomenon using bovine primary cells *in vitro*.

During pathogenic infection, HAD phosphatases may function as effector proteins and secrete proteins to regulate the biological processes of host cells. Another effector, Lem4, was co-localized with host cell lysosomes, participated in the formation of extracellular vesicle structures, and interacted with associated phosphorylated host targets (54). A serine-specific phosphatase *Porphyromonas gingivalis* SerB653 can be secreted into the host cytoplasm to co-localize with P65 and inhibit P65-mediated IL-8 promoter activity, IL-8 expression, and P65 nuclear translocation by dephosphorylating the serine at the 563 site of P65, affecting the immune response of host cells (55). Previous studies have found that Ceg4 characterized HAD phosphatase showing strong activity against p38 MAPK, the human homolog of Hog1, and caused a significant reduction in the signal corresponding to phosphorylated p38, in agreement with our finding that the presence of *Mycoplasma* MbovP0725 protein inside the cell could inhibit pro-inflammatory cytokine production by interacting and hampering the phosphorylation of cellular proteins including p38 and ERK1/2. Moreover, this was compromised by the mutation of the key MbovP0725 active site at the $Asp^{15}$ residue, including the D15A mutant protein showing a weaker anti-inflammatory response and losing the ability to suppress the phosphorylation of P38 and ERK proteins. Therefore, we conclude that D15A plays an important role in the MbovP0725 anti-inflammatory function by reducing the phosphorylation of ERK and P38. However, to the best of our knowledge, the mechanism between HAD-like proteins and their interactive proteins is unclear. Only one previous paper showed that the HAD-like protein Ceg4 from *Legionella pneumophila* can inhibit MAP kinases (Fus3p MAPK) in *S. cerevisiae* through its C-terminal region (37). This finding provides additional support to our results that C-terminal D15A is associated with MbovP0725 phosphorylation function. The binding domain between MbovP0725 and ERK or P38 in this study remains to be investigated in the future.

## Conclusion

This study found a novel secreted serine/threonine phosphatase MbovP0725 belonging to the HAD superfamily of *M. bovis*, which could interact with the host P38 and ERK protein and attenuate the phosphorylation of the MAPK signaling pathway, inhibiting IL-1β, IL-6, and TNF-α mRNA expression. The transcriptomic and metabolomic data suggest that MbovP0725 most likely has a phosphatase effect on the glycolytic pathway and nucleotide metabolism.

## MATERIALS AND METHODS

### Cells, bacterial strains, and their culture

The bovine macrophage cell line BoMac used in this study was kindly provided by Judith R. Stabel from Johne's Disease Research Project (56). BoMac cells were cultured in Roswell Park Memorial Institute 1640 (Hyclone, USA) supplemented with 10% heat-inactivated fetal bovine serum (FBS) (Gibco, USA). The bovine mammary epithelial cell line MAC-T was donated by Professor Mark Hanigan (Virginia Tech University, USA) and kept in this laboratory. The MAC-T cells were cultured in Dulbecco's modified eagle medium/F12 medium (DMEM/F12, Hyclone) supplemented with 10% heat-inactivated FBS (Gibco).

The *M. bovis* wild-type strain HB0801 was isolated from the lung taken from a cow with clinical pneumonia in Hubei province, China, in 2008 (GenBank accession no. NC.018077.1) (6). It was propagated in a PPLO medium (BD Company, USA) at 37℃, incubated under an atmosphere of 5% $CO_2$. The *M. bovis* transposon mutant strain T8.66 was grown similarly in a PPLO medium with 100 μg/mL gentamycin and 10 μg/mL puromycin. An *E. coli* strain DH5α (TransGen, Beijing, China) was grown in Luria–Bertani broth (LB, Oxoid, UK) with proper antibiotics.

## *In silico* identification and sequence analysis of MbovP0725

The amino acid sequences of MbovP0725 from *M. bovis* HB0801 genome (GenBank: AFM52074.1) were obtained from the NCBI database. Protein homologs to MbovP0725 in *Mycoplasma* and other bacteria were identified with the online server hhPred (https://toolkit.tuebingen.mpg.de/tools/hhpred). Homologous sequence alignment was conducted online with CLUSTALW (https://www.genome.jp/tools-bin/clustalw) and ESPript (https://espript.ibcp.fr/ESPript/cgi-bin/ESPript.cgi).

## Expression and purification of recombinant MbovP0725 (rMbovP0725) and its site-directed mutation protein

After codon adaptation for expression in Mbov_0725 genes in *E. coli*, the full length of the modification Mbov_0725 gene was cloned into pET-30a vectors after restriction enzyme digestion by *Xho I* and *BamH I* and T4 ligases as described previously. Site-directed mutagenesis to create the mutants D15A, T48A, K210A, and D236A was carried out using standard polymerase chain reaction (PCR)-based techniques using the primers in Table S1 with a Q5 site-directed mutagenesis kit (NEB, Beijing, China), where the resulting PCR product was inserted into pET-30a vectors. The construct sequences were verified by Sanger sequencing. The recombinant plasmids pET-30a-Mbov_0725 and its mutants were transformed into *E. coli* BL21 competent cells (Vazyme, Nanjing, China) in the LB medium supplemented with 10 µg/mL kanamycin and induced by the addition of 1 mM isopropyl-β-D-thiogalactopyranoside (IPTG) at 16°C overnight. The cultures were harvested by centrifugation, and pellets were lysed by sonication on ice in protein binding buffer. All further purification was conducted at 4°C. The cell lysate was clarified by ultracentrifugation at $10,000 \times g$ for 30 min, and 5 mL of nickel–nitrilotriacetic acid resin was added in a chromatography column (Thermo Fisher Scientific, USA). Resin was washed with binding buffer, and protein was eluted with elution buffer containing 500 mM imidazole. Proteins were further concentrated using an ultrafiltration device, before using Triton X-114 to remove the endotoxin (<0.001 EU/µg protein) as described previously (57). Proteins were quantified using a bicinchoninic acid protein assay kit (Beyotime Biotechnology, Shanghai, China).

The mouse antiserum to rMbovP0725 was prepared previously, and its immunity serum titer was confirmed by western blot (20). The recombinant eukaryotic expression plasmid pEGFP-N1-Mbov_0725 was extracted using an endo-free plasmid kit (Omega Bio-tek, GA, USA). All the above products were verified by sequencing.

## Assays of phosphatase activity

The activity of rMbovP0725 enzymes was measured by monitoring the hydrolysis of *p*NPP (New England BioLabs) to *p*-nitrophenyl as described previously (37). Briefly, reactions consisted of 1.77 mM *p*NPP, 50 mM Tris-HCl (pH 8.0), 10 mM MgCl$_2$, and 30 µg rMbovP0725 at 37°C for 0, 0.25, 0.5, 1, 1.5, 2, 4, and 12 h. Control reactions had no protein. Each reaction was done in triplicate wells. After the reaction was finished, 18 µL 6 M NaOH was added to stop the reaction, and the absorbance was read at 405 nm using a Spectramax plate reader.

For kinetics determinations, *p*NPP and rMbovP0725 were used at the concentrations specified previously. To determine the optimum metal requirement, different metal solutions including 10 mM KCl, NaCl, CaCl$_2$, MnCl$_2$, and MgCl$_2$ were added. In order to determine the optimal pH of the phosphatases, the pH of the buffer was adjusted to 5, 6, 7, 8, 9, and 10.

## Substrate specificity testing

To determine the specificity of rMbovP0725 toward serine, threonine, and tyrosine, a malachite green phosphatase assay kit (Abnova) was used, which used O-phosphoserine (p-Ser) for serine phosphate, O-phosphothreonine (p-Thr) for threonine phosphate, and p-Tyr for tyrosine phosphatase as substrates in a 96-well plate format. The assay was

performed following the manufacturer's instructions, using the buffer and MbovP0725 concentration as described above. Each reaction was repeated three times.

## Analysis of MbovP0725-secreted proteins from *M. bovis*

The *M. bovis* HB0801 was cultured in a PPLO medium for 8, 12, 24, 36, and 48 h. At different times, cells were pelleted by centrifugation at 12,000 rpm for 5 min at 4°C, and the supernatant was carefully aspirated and concentrated into a clean 15-mL ultrafiltration tube and stored at −80°C. The cell pellet was resuspended in 1 mL cold PBS, and SDS-PAGE lysis buffer was added, before being heated at 100°C for 10 min, and the proteins within them were separated by 12% SDS-PAGE. The gel was then stained with Coomassie blue or subjected to western immunoblotting.

## Identification of Mbov_0725 mutant and construction of its complementary strain

The Mbov_0725 knockout mutant T8.66 was identified from the transposon-mediated *M. bovis* mutant library previously constructed in this laboratory (28). The mutated site was at nt 108 of the Mbov_0725 coding sequence or nt 857920 of the *M. bovis* HB0801 genome.

To construct a CT8.66 strain for this mutant T8.66, a DNA fragment containing the Mbov_0725 sequence following the P40 promoter from *M. agalactiae* was synthesized (Beijing Tianyi Huiyuan Bioscience & Technology Inc.). The synthetic DNA fragment was ligated into plasmid pOH/P at the *Xho I* and *BamH I* restriction site to generate plasmid pCP-T8.66. The DNA constructions were verified by DNA sequencing and then transformed into *M. bovis* T8.66 to generate the complementary strain CT8.66 (58). Single colonies were selected with 10 µg/mL puromycin in the medium and confirmed with DNA sequencing. The T8.66 and CT8.66 strains were cultured in a PPLO medium containing 100 µg/mL gentamycin and 10 µg/mL puromycin, respectively, and their growth curves were determined in parallel with a standard plate counting method, and their morphological colony was observed under a microscope (Olympus, SZX16).

The MbovP0725 expression in T8.66 and CT8.66 was evaluated with a western blotting assay. Both strains were cultured in a 20-mL PPLO medium with the necessary antibiotics for 36 h and precipitated by centrifugation at 12,000 × *g* for 10 min. The pellet of each strain was then suspended in 1 mL of PBS and lysed by sonication at 200 W, on ice for 5 min. The proteins in the lysate were then separated with 12% SDS-PAGE and transferred onto a polyvinylidene fluoride (PVDF) membrane (Millipore, Darmstadt, Germany). The membrane was incubated with mouse antiserum (1:500) directed against rMbovP0725 and rMbovP579 previously made by this lab at room temperature for 1 h (59). After the membrane was washed, it was overlaid with the horseradish peroxidase (HRP)-conjugated goat anti-mouse IgG antibody (1:5,000, Southern Biotech) for 1 h at room temperature. The bands on the membrane were then visualized with the WesterBright ECL western blotting detection kit (Advansta, CA, USA).

## Cytokine production induced by *M. bovis* strains and purified rMbovP0725 protein in BoMac and MAC-T cells, respectively

BoMac cells were infected with *M. bovis* HB0801, T8.66, and CT 8.66 at an MOI of 1,000 for 12 h by qRT-PCR. The total cellular RNA was extracted using the TRIzol reagent (Invitrogen, CA, USA) and was reverse-transcribed into cDNA using HiScript Reverse Transcriptase (Vazyme, Nanjing, China). The mRNA levels of IL-1β, IL-6, and TNF-α were detected on a VillA7 real-time PCR system (Applied Biosystems, CA, USA) using SYBR Green Master Mix (Vazyme) and quantified using the $2^{-\Delta\Delta Ct}$ method. The primers used are listed in Table S1.

To investigate the effect of recombinant protein MbovP0725 on cytokine production, BoMac and MAC-T cells were stimulated with rMbovP0725 protein at various concentrations of 1, 5, 10, and 20 µg/mL and 1 µg/mL LPS for 12 h. The cells were then collected to

determine IL-1β, IL-6, and TNF-α mRNA levels, as described above. To further determine the function of endogenous MbovP0725, the BoMac and MAC-T cells were grown to a confluence of 70% at the time of transfection. Cells were transfected with endotoxin-free pEGFP-Mbov_0725 and pEGFP-N1 using the jetPRIME reagent (Polyplus, Illkirch, France) according to the manufacturer's instructions. After 24 h of transfection, cells were then stimulated with LPS for an additional 12 h and then harvested to detect IL-1β, IL-6, and TNF-α mRNA levels as described above.

## *Mycoplasma* adhesion assay

The EBL and MAC-T cells were used for *Mycoplasma* adhesion assays as previously described (58). Cells were seeded in 24-well plates at a density of $10^5$ cells per well. After overnight culture, monolayers were washed with sterile PBS and inoculated with $10^8$ CFU of *Mycoplasma* in PBS. At indicated time points of 30, 60, and 120 min incubation at 37℃, non-adherent mycoplasmas were removed by washing with PBS, and cell monolayers were lysed with 1 mL ddH$_2$O. Adherent mycoplasmas were counted by plating serial dilutions onto a solid PPLO medium, and each sample was tested in triplicate.

## Transcriptomic analysis

The *M. bovis* HB0801 and T8.66 cells were grown to the mid-logarithmic phase in a PPLO medium as described above. Three independent biological replicates of bacteria pellets were collected in each group, total RNA was extracted separately by TRIzol, and the quality of the RNA samples was examined using the 2100 bioanalyzer (Agilent, USA). Library construction and Illumina sequencing were performed at Novogene Bioinformatics Technology Co., Ltd (Beijing, China). An RNA-seq analysis was performed according to the protocol recommended by the manufacturer (Illumina Inc., UK). The reads from different samples were mapped to the whole-genome assembly using Bowtie 2-2.3.4.3. HTSeq 0.9.1 was used to count the read numbers mapped to each gene, and the reads per kilobase per million reads of each gene were calculated based on the length of the gene and read counts mapped to this gene.

A differential expression analysis of HB0801 and T8.66 cells of each group with three biological replicates was performed using the DESeq2 R package (1.20). Genes with an adjusted *P*-value <0.05 and |log2(fold change)|>0 found by DESeq were assigned as differentially expressed. Clusters of orthologous groups of protein classification and Pfam domain assignment were conducted on the DEGs. The KOBAS software was used to test the statistical enrichment of DEGs in KEGG pathways. The transcriptomic sequence data have been deposited in the NCBI databases (accession to cite for these SRA data: PRJNA972635).

## Transcriptome data validation

Among all DEGs, 18 genes identified by RNA-seq analysis were selected for validation by qRT-PCR (primers are listed in Table S1). The sample preparation procedure was the same as described above. Total RNA was extracted using TRIzol, and then, cDNA was synthesized from total RNA using the PrimeScript II first-strand cDNA synthesis Kit (Vazyme, China) according to the manufacturer's instructions.

All qRT-PCR experiments were performed in three technical replicates using 16s as the reference gene. The qRT-PCR primers used in this study are described in Table S1. The cycle conditions were 95℃ for 5 min, 40 cycles of 95℃ for 10 s, 60℃ for 10 s, and 72℃ for 15 s, and the melting curve temperature was 72℃ to 95℃. Gene expression was calculated using the $2^{-\Delta\Delta Ct}$ relative expression method.

## Untargeted metabolomic analysis

We cultured 10 mL *M. bovis* WT HB0801 and T8.66 cells in a PPLO medium to the log phase and flash-froze them in liquid nitrogen until metabolomic analysis. Six independent biological replicates of bacteria pellets were collected in each group. Samples

were thawed on ice and subjected to ultrasonication at room temperature for 30 min, subsequently placed on ice for 30 min, and centrifuged at 12,000 rpm at 4°C for 10 min. Then, 850 µL of the supernatant was taken and dried in a vacuum concentrator. We added 200 µL of 2-chlorobenzalanine solution made with 50% acetonitrile solution to redissolve the samples, which were subsequently filtered through a 0.22-µm membrane, with 20 µL of the filtrate from each sample pooled into a quality control sample to be used for data normalization. The remaining samples were used for LC-MS detection.

## LC-MS/MS analysis

The research used LC-MS/MS technology for untargeted metabolomic analysis, using a high-resolution Q Exactive mass spectrometer (Thermo Fisher Scientific, USA) to collect data from both positive and negative ions to improve metabolite coverage. Data pre-processing, statistical analysis, metabolite classification annotations, and functional annotations were performed using the self-developed metabolomic R package metaX and the metabolome bioinformatic analysis pipeline. The multivariate raw data are dimensionally reduced by PCA to analyze the grouping, trends, and outliers of the observed variables in the data set. PLS-DA, VIP values of the first two principal components of the model, combined with the variability analysis, fold change, and Student's test were used to screen for differential metabolites. Significantly differential metabolite screening conditions included the VIP of the first two principal components of the PLS-DA model exceeding 1 and 2, fold change exceeding 1.2 or less than 0.83, and $q$-value less than 0.05.

## Western blot assays

For exogenous rMbovP0725 protein treatment, MAC-T cells were stimulated with 1, 5, 10, and 20 µg/mL of endotoxin-removal rMbovP0725 and 1 µg/mL LPS for 2 h. Cells treated with PBS were used as a negative control. Endogenous MbovP0725 was also produced by the transfection of pEGFP-MbovP0725 into MAC-T cells. After MbovP0725 treatment, the cell was lysed using RIPA lysis buffer (Sigma-Aldrich) containing inhibitors of protease and phosphatase (Roche, Basel, Switzerland), and the lysates were then centrifuged at 12,000 × $g$, 4°C for 10 min to harvest the supernatant. Proteins from the supernatant were separated by 12% SDS-PAGE and transferred to PVDF membranes, which were blocked with 5% skim milk in TBST. The membranes were incubated overnight with antibodies against non-phosphorylated and phosphorylated ERK1/2 and P38 (Cell Signaling Technology) at 1:1,000 dilution in TBST at 4°C. After washing, membranes were incubated with anti-rabbit or mouse HRP for 1 h at room temperature. Blots were developed with WesternBright ECL (Advansta, CA, USA) and visualized using a chemiluminescence instrument (Tanon, Shanghai, China). Gray value analysis was developed using ImageJ.

## Immunoprecipitation-MS assay to verify p38 and ERK binding to MbovP0725

Briefly, the HEK293T cells were cultured and then transfected using the jetPRIME reagent with endotoxin-free pCAGGS-HA-Mbov_0725. Cells transfected with pCAGGS-HA and pCAGGS-HA-Mbov_0725 D15A were used as control. After 24 h of incubation, the cells were washed once by cold PBS and lysed by NP40 lysis buffer (Beyotime, Shanghai, China) containing inhibitor of protease on ice for 40 min. The whole cell lysates were incubated with 2 µg anti-HA antibody overnight at 4°C, and then, 50 µL of Protein A/G agarose beads was added and incubated for another 4 h. The beads were centrifugated and washed five times with NP40 buffer and then eluted with SDT buffer consisting of 4% SDS, 100 mM Tris/HCl, and 1 mM DTT, at a pH of 7.6. A western blotting assay was then performed, and the proteins were immunodetected with antibodies directed against HA, ERK1/2, and P38 antibodies.

## Statistical analysis

Data are expressed as the mean ± SD. Student's $t$-test was used for single comparisons and one-way or two-way ANOVA for multiple comparisons with GraphPad Prism version 8 software (GraphPad Software, La Jolla, CA, USA). Statistical significance is expressed at the following four levels: not significant (ns, $P > 0.05$), $^{*}P < 0.05$, $^{**}P < 0.01$, and $^{***}P < 0.001$.

## ACKNOWLEDGMENTS

This research was funded by the Projects of Youth program of the National Natural Science Foundation of China (#32002290), National Natural Science Foundation of International (Regional) Cooperation Projects (#31661143015), earmarked fund for China Agriculture Research System (Beef/yaks) (#CARS-37), Southwest Minzu University Research Startup Funds (RQD2023031/16011231007), and Fundamental Research Funds for the Central Universities, Southwest Minzu University (ZYN2023045).

## AUTHOR AFFILIATIONS

[1]College of Animal & Veterinary Sciences, Key Laboratory of Animal Medicine of Sichuan Province, Southwest Minzu University, Chengdu, China

[2]The State Key Laboratory of Agricultural Microbiology, College of Veterinary Medicine, Hubei Hongshan Laboratory, Huazhong Agricultural University, Wuhan, China

[3]Key Laboratory of Development of Veterinary Diagnostic Products, Ministry of Agriculture and Rural Affairs, Huazhong Agricultural University, Wuhan, China

[4]Hubei International Scientific and Technological Cooperation Base of Veterinary Epidemiology, Huazhong Agricultural University, Wuhan, China

[5]Key Laboratory of Ruminant Bio-products of Ministry of Agriculture and Rural Affairs, Huazhong Agriculture University, Wuhan, China

[6]International Research Center for Animal Disease, Ministry of Science and Technology, Huazhong Agricultural University, Wuhan, China

## AUTHOR ORCIDs

Hui Zhang http://orcid.org/0000-0002-9491-4561
Aizhen Guo http://orcid.org/0000-0002-7460-8356

## FUNDING

| Funder | Grant(s) | Author(s) |
|---|---|---|
| MOST \| National Natural Science Foundation of China (NSFC) | 32002290 | Hui Zhang |
| National natural science foundation of international (Regional) Cooperation Projects | 31661143015 | Aizhen Guo |
| China Agricultural Research System (CARS) | CARS-37 | Aizhen Guo |
| Research startup funds | RQD2023031/16011231007 | Hui Zhang |
| Fundamental research funds for the central unvieristies Southwest MInzu University | ZYN2023045 | Hui Zhang |

## AUTHOR CONTRIBUTIONS

Hui Zhang, Conceptualization, Data curation, Validation, Writing – original draft | Yiqiu Zhang, Data curation, Formal analysis | Doukun Lu, software, Validation | Xi Chen, Supervision | Yingyu Chen, investigation | Changmin Hu, Supervision, Writing – review and editing | Aizhen Guo, Funding acquisition, Project administration, Resources, Supervision, Writing – review and editing

## ADDITIONAL FILES

The following material is available online.

## Supplemental Material

**Table S1 (mSystems00891-23-s0001.docx).** Primers.

## Open Peer Review

**PEER REVIEW HISTORY (review-history.pdf).** An accounting of the reviewer comments and feedback.

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
