## [Reviewer comments · mSystems]

MbovP0725, a secreted serine/threonine phosphatase inhibits the host inflammatory response and affects metabolism in *Mycoplasma bovis*

zhang hui, Yiqiu Zhang, Doukun Lu, Xi Chen, Yingyu Chen, Changmin Hu, and Aizhen Guo

Corresponding Author(s): Aizhen Guo, Huazhong Agricultural University

Review Timeline:

Submission Date:	August 23, 2023
Editorial Decision:	October 18, 2023
Revision Received:	December 15, 2023
Accepted:	January 17, 2024

Editor: Tricia Van Laar

Reviewer(s): The reviewers have opted to remain anonymous.

Transaction Report:

DOI: <https://doi.org/10.1128/msystems.00891-23>

October 18, 2023

Prof. Aizhen Guo
College of Veterinary Medicine
Huazhong Agricultural University, Wuhan
Shizishan 1
Wuhan, Hubei 430070
China

Re: mSystems00891-23 (Metabolic characteristics and pathogenicity of a secreted serine/threonine phosphatase encoded by MbovP0725 in *Mycoplasma bovis*)

Dear Prof. Aizhen Guo:

Thank you for submitting your manuscript to mSystems. We have completed our review and I am pleased to inform you that, in principle, we expect to accept it for publication in mSystems. However, acceptance will not be final until you have adequately addressed the reviewer comments.

Please ensure you include the required "Data Availability" paragraph at the end of the materials and methods.

Preparing Revision Guidelines

Please return the manuscript within 60 days; if you cannot complete the modification within this time period, please contact me. If you do not wish to modify the manuscript and prefer to submit it to another journal, please notify me of your decision immediately so that the manuscript may be formally withdrawn from consideration by mSystems.

Sincerely,

Tricia Van Laar

Editor, mSystems

Journals Department
Reviewer comments:

Reviewer #1 (Comments for the Author):

Hui Zhang, Doukun Lu and co-workers have thoroughly investigated the role of MbovP0725 as a secreted serine/threonine phosphatase of *Mycoplasma bovis* that contributes to pathogenesis using a wide range of approaches and techniques. The subject is interesting and relevant to the microbiology community. The authors have done a nice job, both in quality and quantity of performed work, and there is obvious novelty. The paper has clear goals using relevant experiments, and conclusions are generally well supported by the results. I have some issues that should be addressed.

General remarks

- (1) The authors found that MbovP0725 could interact with the host P38 and ERK proteins and attenuate the phosphorylation of these two proteins, which inhibits the host pro-inflammatory response induced by LPS. This is a very interesting research result and well stated. But little information on how the protein down-regulates the phosphorylation of P38 and ERK proteins, does the D15A amino-acid of MbovP0725 influence the phosphorylation function? And how?
- (2) The authors then performed transcriptomic and metabolomic analysis to identify the differentially-expressed genes and metabolites from *M. bovis* WT and T8.66 strains (MbovP0725 gene deleted), where some ribo- and deoxyribo-mononucleotides products and differentially expressed genes responsible for glycolysis and the *pta-ackA* pathway were detected. However, the authors didn't explore the role of these differential proteins and metabolites in inhibiting the host inflammatory responses. Hence, the correlation between *M. bovis* transcriptomic/metabolomic analyses and the protein of MbovP0725's inhibition of host cell inflammatory response is relatively weak. As the manuscript comprises two distinct parts, it is advisable for the authors to consider revising the title as, for example, MbovP0725, a secreted serine/threonine phosphatase inhibits the host inflammatory responses and affects metabolic of *Mycoplasma bovis*.
- (3) In L475-478, I am puzzled as to why the authors chose to employ transcriptomic and metabolomic analyses instead of phosphoproteomics to investigate whether MbovP0725 influences the phosphorylation status of other *M. bovis* proteins. As energy metabolism plays a role in various physiological functions, this does not serve as direct evidence for the MbovP0725's role of phosphorylation.

Results

- (1) Figure 1 D-I (L593-94) lacks negative controls. It is advisable to use some proteins such as BSA or other non-phosphatase *M. bovis* proteins as negative controls, rather than using 'no protein'.
- (2) If Figure 1 D is used as the standard, please provide supplementary information in the Figure legend regarding the values of rMbovP0725 protein concentration, pH ion concentration, and temperature.
- (3) What are the 'control' in Figure 7 and the 'NC' in Figure 8? Please describe it in the legend.
- (4) The D15A, T48A, K210A and D256A mutations prevented phosphatase activity of rMbovP0725 in Figure 2D. Why was only the D15A mutant protein tested without conducting experiments on other mutant proteins in Figure 9?
- (5) In Figure 10A, it is suggested to include a negative control group with LPS + other proteins.

Reviewer #2 (Comments for the Author):

This article found *Mycoplasma bovis* MbovP0725 is a secreted serine/threonine phosphatase that suppresses inflammatory response via MAPK p38 and ERK. It is very importance to understand the mechanism of suppress immune response of *M. bovis*. Here are some questions:

Line 157: eighteen 18? Is it written wrong?

1. Llin 164: from 12 until 48h? that is not correct enough. Several English expressive errors have been found in this article, and it is recommended to improve the English writing
2. Figure 3A, compared to WT, less Mbovp0725 is expressed, D,E, most results showed statistical significance of differences between T8.66 and CT8.66, not between WT and T8.66.
3. In this article, MbovP0725 was deleted from wildtype (WT) *M. bovis* HB0801 (T8.66), is there any possible to construct a MbovP0725 over expressive strain to further indicate that MbovP0725 can suppress inflammatory response?
4. In Figure 3, 7, 8, the production of IL-1 β , IL-6 and TNF- α were detected at mRNA levels, did you detect them at protein expression levels?

Hui Zhang, Doukun Lu and co-workers have thoroughly investigated the role of MbovP0725 as a secreted serine/threonine phosphatase of *Mycoplasma bovis* that contributes to pathogenesis using a wide range of approaches and techniques. The subject is interesting and relevant to the microbiology community. The authors have done a nice job, both in quality and quantity of performed work, and there is obvious novelty. The paper has clear goals using relevant experiments, and conclusions are generally well supported by the results. I have some issues that should be addressed.

General remarks

- (1) The authors found that MbovP0725 could interact with the host P38 and ERK proteins and attenuate the phosphorylation of these two proteins, which inhibits the host pro-inflammatory response induced by LPS. This is a very interesting research result and well stated. But little information on how the protein down-regulates the phosphorylation of P38 and ERK proteins, does the D15A amino-acid of MbovP0725 influence the phosphorylation function? And how?
- (2) The authors then performed transcriptomic and metabolomic analysis to identify the differentially-expressed genes and metabolites from *M. bovis* WT and T8.66 strains (MbovP0725 gene deleted), where some ribo- and deoxyribo-mononucleotides products and differentially expressed genes responsible for glycolysis and the *pta-ackA* pathway were detected. However, the authors didn't explore the role of these differential proteins and metabolites in inhibiting the host inflammatory responses. Hence, the correlation between *M. bovis* transcriptomic/metabolomic analyses and the protein of MbovP0725's inhibition of host cell inflammatory response is relatively weak. As the manuscript comprises two distinct parts, it is advisable for the authors to consider revising the title as, for example, MbovP0725, a secreted serine/threonine phosphatase inhibits the host inflammatory responses and affects metabolic of *Mycoplasma bovis*.
- (3) In L475-478, I am puzzled as to why the authors chose to employ transcriptomic and metabolomic analyses instead of phosphoproteomics to investigate whether MbovP0725 influences the phosphorylation status of other *M. bovis* proteins. As energy metabolism plays a role in various physiological functions, this does not serve as direct evidence for the MbovP0725's role of phosphorylation.

Results

- (1) Figure 1 D-I (L593-94) lacks negative controls. It is advisable to use some proteins such as BSA or other non-phosphatase *M. bovis* proteins as negative controls, rather than using 'no protein'.
- (2) If Figure 1 D is used as the standard, please provide supplementary information in the Figure legend regarding the values of rMbovP0725 protein concentration, pH ion concentration, and temperature.
- (3) What are the 'control' in Figure 7 and the 'NC' in Figure 8? Please describe it in the legend.
- (4) The D15A, T48A, K210A and D256A mutations prevented phosphatase activity of rMbovP0725 in Figure 2D. Why was only the D15A mutant protein tested without conducting experiments on other mutant proteins in Figure 9?
- (5) In Figure 10A, it is suggested to include a negative control group with LPS + other proteins.

Response to editors and reviewers

Dear editor and reviewers,

We would like to thank you for your meticulous reading, insightful comments, and valuable suggestions, which have significantly improved the quality of our manuscript. We have thoroughly considered all the feedback provided by the reviewers and made appropriate revisions accordingly. Moreover, we performed extra experiment to show MbovP0725 with D15A mutation lost the ability to suppress phosphorylation of P38 and ERK proteins. This manuscript was edited for proper English language, grammar, punctuation, spelling, and overall style by International Science Editing company and confirmed by a certificate of Editing.

Additionally, we kindly request a change in author order as reflected in the revised manuscript. This adjustment is based on Yiqiu Zhang's substantial contribution during the experiment. All authors are in agreement with this reordering decision. We firmly believe that our response adequately addressed all concerns raised by the reviewers and sincerely hope that our revised manuscript will meet your publication standards.

Best wishes,

All authors.

Reviewer #1 (Comments for the Author):

Hui Zhang, Doukun Lu and co-workers have thoroughly investigated the role of MbovP0725 as a secreted serine/threonine phosphatase of *Mycoplasma bovis* that contributes to pathogenesis using a wide range of approaches and techniques. The subject is interesting and relevant to the microbiology community. The authors have done a nice job, both in quality and quantity of performed work, and there is obvious novelty. The paper has clear goals using relevant experiments, and conclusions are generally well supported by the results. I have some issues that should be addressed.

General remarks

(1) The authors found that MbovP0725 could interact with the host P38 and ERK proteins and attenuate the phosphorylation of these two proteins, which inhibits the host pro-inflammatory response induced by LPS. This is a very interesting research result and well stated. But little information on how the protein down-regulates the phosphorylation of P38 and ERK proteins, does the D15A amino-acid of MbovP0725 influence the phosphorylation function? And how?

Response: Thank you very much for your critical and kind comments.

Firstly, we predicted with bioinformatical assay phosphorylation sites within catalytic residues at 15, 48, 210 and 236 in MbovP0725. Then we performed alanine substitution at 15, 48, 210 and 236 sites in MbovP0725 and two alanine substitutions at residues 48, 210 significantly decreased, while another two alanine substitutions at 15, 236 abolishes its phosphatase activity indicating 15, 236 residues mostly play an important role in regulating its phosphorylation process. Further to answer your question about whether the mutation D15A influence phosphorylation of P38 and ERK proteins, we performed extra western blotting assay for D15A mutant and showed that MbovP0725 with D15A mutation lost the ability to suppress phosphorylation of P38 and ERK proteins (this extra result shown in the followed figure has been integrated into Fig.10 in the revised MS). Accordingly, compared to wild-type MbovP0725(rMbovP0725), the D15A mutant protein showed a weaker anti-inflammatory

response (shown in Figure 9). Therefore, we concluded that D15A plays an important role in MbovP0725 anti-inflammatory function by down-phosphorylation of ERK and P38. However, to our best knowledge, the mechanism between HAD-like proteins and their interactive proteins is unclear. Only one previous paper showed that HAD-like protein Ceg4 from *Legionella pneumophila* can inhibit MAP kinases (Fus3p MAPK) in *S. cerevisiae* through its C-terminal region (Quaile et al., 2018). This finding provides additional support to our results that C-terminal D15A is associated with MbovP0725 phosphorylation function. The binding domain between MbovP0725 and ERK or P38 in this study remains investigation in the future. We added this reference and discussion to explain the possibility and limitation in the discussion in the revised version (please see line 515-520, 657-669).

The mutant D15A protein did not effect on the phosphorylation of MAPK P38 and ERK. A. Western blot analysis was performed to assess the activation and phosphorylation of MAPK P38 and ERK1/2 in MAC-T cells incubated with different doses of the mutant protein, stimulated with LPS for 2 h. B. Densitometry qualification normalized the phosphorylated P38 and ERK1/2 levels by total P38 and ERK (no asterisk means no significant difference).

(2) The authors then performed transcriptomic and metabolomic analysis to identify the differentially-expressed genes and metabolites from *M. bovis* WT and T8.66 strains (MbovP0725 gene defected), where some ribo- and deoxyribo-mononucleotides products and differentially expressed genes responsible for glycolysis and the *pta-ackA* pathway were detected. However, the authors didn't explore the role of these differential proteins and metabolites in inhibiting the host inflammatory responses. Hence, the correlation between *M. bovis* transcriptomic/metabolomic analyses and the protein of MbovP0725's inhibition of host cell inflammatory response is relatively weak. As the manuscript comprises two distinct parts, it is advisable for the authors to consider revising the title as, for example, MbovP0725, a secreted serine/threonine phosphatase inhibits the host inflammatory responses and affects metabolic of *Mycoplasma bovis*.

Response: Thank you for your valuable comments and we agree with your comments. As you suggested, we added the discussion on the correlation between *M. bovis* transcriptomic/metabolomic analyses and MbovP0725's inhibition of host cell inflammatory response in Discussion section as follows :

In our study, among differential genes revealed by transcriptomic analyses, 15 differential genes were enriched in the glycolytic and *Pta/AckA* pathways. As previously reported, many glycolytic enzymes in mycoplasma are localized on the cell surface and act as extracellular matrix-binding

proteins involved in cytoadherence and modulating the host immune system. *Mycoplasma gallisepticum* *Pdha/Pdhb* are reported to be adhesions and surface-exposed immunogenic proteins (Qi et al., 2018). Eight Glycolytic enzymes in *M. pneumoniae* including *Pdh-A,B,C*, *GapA*, *Ldh*, *Pgm*, *Pyk*, *Tkt* were confirmed as surface expressed and able to interact with plasminogen associated with adhesion and invasion (Grundel et al., 2016). Additionally, most of these proteins were metabolic enzymes involved in carbohydrate and energy metabolism. These glycolytic genes have been previously reported to be phosphorylated in *M. pneumoniae* for ATP generation (Schmidl et al., 2010). Moreover, *Pta-AckA* pathway plays important role in bacterial fitness through the control of central metabolism to sustain balanced growth and cellular homeostasis (Schütze et al., 2020). In *Staphylococcus aureus*, inactivation of *Pta-AckA* pathway increased the rate of glucose consumption, led to a metabolic block at the pyruvate node and enhanced carbon flux through glycolysis. Therefore, we speculate that the up-regulated genes in glycolytic and down-regulated genes in *pta/AckA* pathways might supply more ATP generation for intracellular survival of *M. bovis*, which might be induce stronger anti-inflammatory effect contribute to immune escape.

Meanwhile, , 20 differential metabolites with high VIP scores by metabolomic analyses were enriched in nucleotide metabolism, amino acids metabolism and carbon metabolism, etc. Furthermore, we observed an accumulation of two metabolites 9-hodes and 4-phenylbutyric acid with most differentiation in the mutant strain of MbovP0725. The former is found to oxidize free fatty acids in low-density lipoproteins; while the later is a derivative of butyric acid produced through bacterial fermentation processes. Previous studies demonstrated that 9-hodes acts as a pro-inflammatory mediator through activation of GPR132 receptors on monocytes and macrophages (Vangaveti et al., 2010). In addition, 4-phenylbutyric acid has been shown to inhibit ER stress and reduce inflammatory mediators expression (Choi et al., 2021).

Interestingly, some of glycolytic enzymes have multiple functions in metabolism: eg. The glycolytic kinase is also active in nucleotide phosphorylation thus replacing the nucleoside diphosphate kinase(Dutow et al., 2010). *M. pneumoniae* possesses a lactate dehydrogenase alternatively the NAD oxidase may directly oxidize NADH₂ for the regeneration of NAD (Halbedel et al., 2007). Although both differential genes and metabolites are involved in glycolytic pathway and ATP generation, their metabolic pathways function and inflammatory response in *M. bovis* remain undefined requiring further exploration.

Therefore, we added the references to explain this phenomenon in the discussion (please see line 535-553, 590-604).

As suggested, we corrected the title as “MbovP0725, a secreted serine/threonine phosphatase inhibits the host inflammatory response and affects metabolism in *Mycoplasma bovis*”.

(3) In L475-478, I am puzzled as to why the authors chose to employ transcriptomic and metabolomic analyses instead of phosphoproteomics to investigate whether MbovP0725 influences the phosphorylation status of other *M. bovis* proteins. As energy metabolism plays a role in various physiological functions, this does not serve as direct evidence for the MbovP0725's role of phosphorylation.

Response: Thank you for your critical comment.

We agree with you that energy metabolism plays a role in various physiological functions, and phosphoproteomics could provide direct evidence for the MbovP0725's role of phosphorylation. In fact,

we attempted to compare the phosphorylation patterns between *M. bovis* WT and T8.66 using 1D electrophoresis and Pro-Q diamond-stained gels and didn't find differentially phosphorylated bands shown as follows and then stopped further investigation of phosphoproteomics. In the future, it is worthwhile to use more sensitive phosphoproteomics methods to explore the MbovP0725's role of phosphorylation.

To more accurately express this idea, we corrected the sentence in L475-478 as follows “We speculate that MbovP0725, as a phosphatase, may alter the metabolism and pathogenicity of *M. bovis*. To determine this, we performed transcriptomic and metabolomic analyses to identify the differential genes and metabolites between *M. bovis* WT and T8.66 strains”.

Figure. The image of phosphorylated proteins of *M. bovis* wildtype stain HB0801, MbovP0725 mutant strain T8.66 and its complementary stain CT8.66. Bands with high Pro-Q Diamond staining representing phosphorylated proteins.

Results

(1) Figure 1 D-I (L593-94) lacks negative controls. It is advisable to use some proteins such as BSA or other non-phosphatase *M. bovis* proteins as negative controls, rather than using 'no protein'.

Response: Thank you for your comment. We agree with your suggestion. It would be ideal to add some proteins such as BSA or other non-phosphatase *M. bovis* proteins as negative controls. We are very sorry for this omission. Alternatively, please allow us to borrow the evidence that the different mutated proteins showed no or significantly decreased phosphatase activity to support the conclusion in this experiment (Figure 2D).

(2) If Figure 1 D is used as the standard, please provide supplementary information in the Figure legend regarding the values of rMbovP0725 protein concentration, pH ion concentration, and temperature.

Response: Thank you for your thoughtful comment. We have added additional details regarding Figure 1 legend as follows (please see line 187-204 in the revised manuscript): “D. Phosphatase activity of rMbovP0725 (30 μ g) with pNPP substrate at various time points (0 h, 0.25 h, 0.5 h, 1 h, 1.5 h, 2 h, 4 h, 12 h) in presence of 5 mM Mg^{2+} . E. Phosphatase activity of rMbovP0725 at different concentrations (0 μ g, 5 μ g, 10 μ g, 20 μ g, 30 μ g, 50 μ g). F. Effect of different monovalent and divalent cation (K^+ , Na^+ , Ca^{2+} , Mn^{2+} , Mg^{2+}) on the phosphatase activity of rMbovP0725. G. Effect of different concentration of Mg^{2+} (0.0625 mM, 0.125 mM, 0.25 mM, 0.5 mM, 1 mM, 5 mM, 10 mM) on the phosphatase activity of rMbovP0725. H. Effect of different pH values (5, 6, 7, 8, 9, 10) on the phosphatase activity of rMbovP0725. I. Effect of different temperature (4 $^{\circ}C$, 16 $^{\circ}C$, 30 $^{\circ}C$, 37 $^{\circ}C$, 42 $^{\circ}C$, 55 $^{\circ}C$, 65 $^{\circ}C$) on

the phosphatase activity of rMbovP0725. Without rMbovP0725 or PNPP were used as a negative control. Values represent Mean \pm SD.”

(3) What are the 'control' in Figure 7 and the 'NC' in Figure 8? Please describe it in the legend.

Response: Thank you for your careful comment. In Figure 7, “Control” indicates no treatment with LPS and “NC” in Figure 8 means PBS treated cells without either LPS or MbovP0725. We have added this information in the legend. Please see line 394 and line 409-410 in our revised manuscript.

(4) The D15A, T48A, K210A and D256A mutations prevented phosphatase activity of rMbovP0725 in Figure 2D. Why was only the D15A mutant protein tested without conducting experiments on other mutant proteins in Figure 9?

Response: Thank you for your critical comments. In our study, we expressed and purified four mutant proteins of MbovP0725. Two of these mutants (T48A and K210A) showed weak phosphatase activity, while the other two mutants (D15A and D236A) showed no detectable activity with *pNPP* (see original manuscript Figure 2). During the process of expression and purification, we observed partial degradation of the D236A mutant suggesting that this mutation affected protein stability. Therefore, we finally selected the D15A mutant protein for further study. We added this explanation in the Discussion section. Please see line 520-523 in our revised manuscript.

(5) In Figure 10A, it is suggested to include a negative control group with LPS + other proteins.

Response: Thank you for your important suggestion. We have performed an additional western blot analysis on LPS + the D15A mutant with different concentrations served as dual controls, one is the negative control group with LPS + other proteins as suggested, the other is control that showed the residue D15 is critical to MbovP0725 phosphorylation function. This extra result has been integrated into Fig. 10 (Fig.10 B, E, F in the revised) in revised version of MS.

We greatly appreciated the reviewer’s valuable comments and suggestions, which have significantly improved the quality of our manuscript.

Reference

- [1]. Choi, Y., Lee, E.G., Jeong, J.H., and Yoo, W.H. (2021). 4-Phenylbutyric acid, a potent endoplasmic reticulum stress inhibitor, attenuates the severity of collagen-induced arthritis in mice via inhibition of proliferation and inflammatory responses of synovial fibroblasts. *The Kaohsiung Journal of Medical Sciences* 37, 604-615.
- [2]. Dutow, P., Schmidl, S.R., Ridderbusch, M., and Stülke, J. (2010). Interactions between Glycolytic Enzymes of *Mycoplasma pneumoniae*. *Microbial Physiology* 19, 134-139.
- [3]. Grundel, A., Jacobs, E., and Dumke, R. (2016). Interactions of surface-displayed glycolytic enzymes of *Mycoplasma pneumoniae* with components of the human extracellular matrix. *Int J Med Microbiol* 306, 675-685.
- [4]. Halbedel, S., Eilers, H., Jonas, B., Busse, J., Hecker, M., Engelmann, S., and Stülke, J. (2007). Transcription in *Mycoplasma pneumoniae*: Analysis of the Promoters of the *ackA* and *ldh* Genes.

Journal of Molecular Biology 371, 596-607.

- [5]. Qi, J.J., Zhang, F.Q., Wang, Y., Liu, T., Tan, L., Wang, S.H., Tian, M.X., Li, T., Wang, X.L., Ding, C., and Yu, S.Q. (2018). Characterization of pyruvate dehydrogenase alpha and beta subunits and their roles in cytoadherence. *Plos One* 13.
- [6]. Quaile, A.T., Stogios, P.J., Egorova, O., Evdokimova, E., Valleau, D., Nocek, B., Kompella, P.S., Peisajovich, S., Yakunin, A.F., Ensminger, A.W., and Savchenko, A. (2018). The Legionella pneumophila effector Ceg4 is a phosphotyrosine phosphatase that attenuates activation of eukaryotic MAPK pathways. *J Biol Chem* 293, 3307-3320.
- [7]. Schmidl, S.R., Gronau, K., Pietack, N., Hecker, M., Becher, D., and Stulke, J. (2010). The phosphoproteome of the minimal bacterium Mycoplasma pneumoniae: analysis of the complete known Ser/Thr kinome suggests the existence of novel kinases. *Mol Cell Proteomics* 9, 1228-1242.
- [8]. Schütze, A., Benndorf, D., Püttker, S., Kohrs, F., and Bettenbrock, K. (2020). The Impact of ackA, pta, and ackA-pta Mutations on Growth, Gene Expression and Protein Acetylation in Escherichia coli K-12. *Frontiers in Microbiology* 11.
- [9]. Vangaveti, V., Baune, B.T., and Kennedy, R.L. (2010). Hydroxyoctadecadienoic acids: novel regulators of macrophage differentiation and atherogenesis. *Ther Adv Endocrinol Metab* 1, 51-60.

Reviewer #2 (Comments for the Author):

This article found Mycoplasma bovis MbovP0725 is a secreted serine/threonine phosphatase that suppresses inflammatory response via MAPK p38 and ERK. It is very importance to understand the mechanism of suppress immune response of M. bovis. Here are some questions:

Line 157: eighteen 18? Is it written wrong?

Response: We apologize for our incorrect writing that caused misunderstanding. We have corrected the wrong number of residues mentioned in line 154 in the revised manuscript.

Line 164: from 12 until 48h? that is not correct enough. Several English expressive errors have been found in this article, and it is recommended to improve the English writing

Response: We are very sorry for our incorrect writing. “until” has been changed to “to”. The revised manuscript has been thoroughly edited by International Science Editing to meet the journal’s standard. Thank you so much for your useful comments.

2. Figure 3A, compared to WT, less Mbovp0725 is expressed, D,E, most results showed statistical significance of differences between T8.66 and CT8.66, not between WT and T8.66.

Response: Thank you for pointing out this problem in our manuscript, and we apologize for our negligent mistake. Actually, we conducted this experiment two times (Figure A and B as below) and observed consistent trends of statistical significance of differences in mRNA levels between T8.66 and CT8.66, and between WT and T8.66 in the two experiments shown in the followed figures. Due to our carelessness, we presented the figure A without statistical significance of differences between WT and T8.66 probably because of the large error bars in this time. We have replaced the original Figure 3 with

a new figure (Figure B as below) clearly demonstrating significant difference between T8.66 and CT8.66, and between WT and T8.66.

Figure. Disruption of MbovP0725 could enhance the mRNA expression of pro-inflammatory cytokines IL-1 β , IL-6 and TNF- α in response to BoMac infected with different strains, as assessed by qRT-PCR. Two-way ANOVA was used to determine the statistical significance of differences between the treatments. * $p < 0.05$, ** $p < 0.01$ and *** $p < 0.001$ indicate statistically significant differences.

3. In this article, MbovP0725 was deleted from wildtype (WT) *M. bovis* HB0801 (T8.66), is there any possible to construct a MbovP0725 over expressive strain to further indicate that MbovP0725 can suppress inflammatory response?

Response: Thank you for your thoughtful comments. We have indeed constructed an overexpressing strain of MbovP0725 in *M. bovis* HB0801 (Please see followed figures). However, its anti-inflammatory response was not stronger as we had anticipated probably the overexpressed MbovP0725 is insufficient to produce significantly stronger inhibitory effect. Therefore we didn't show this result in the current manuscript.

Figure. Overexpression of MbovP0725 in *M. bovis* detected through western blotting assay using polyclonal antibodies against MbovP0725. MbovP0579 was set as control. P1: *M. bovis* HB0801, 1-4: Overexpression strains of MbovP0725 in *M. bovis* HB0801.

4. In Figure 3, 7, 8, the production of IL-1 β , IL-6 and TNF- α were detected at mRNA levels, did you detect them at protein expression levels?

Response: We acknowledge that this is a significant limitation of our study; however, currently it is difficult to detect pro-inflammatory cytokine in BoMac via ELISA and western blotting assay due to their low release levels making detection difficult. Additionally, there is few sensitive commercial ELISA kits for bovine cytokine detection. Therefore, we speculate less IL-1 β , IL-6 and TNF- α secreted probably because of the limited in BoMac, MAC-T cells and insensitive ELISA kits. Further study will be needed to prove the phenomenon using bovine primary cells *in vitro*. We have included this limitation in the Discussion section (line 568-574).

We greatly appreciated the reviewer's valuable comments and suggestions, which have significantly improved the quality of our manuscript.

Re: mSystems00891-23R1 (MbovP0725, a secreted serine/threonine phosphatase inhibits the host inflammatory response and affects metabolism in *Mycoplasma bovis*)

Dear Prof. Aizhen Guo:

Your manuscript has been accepted, and I am forwarding it to the ASM production staff for publication. Your paper will first be checked to make sure all elements meet the technical requirements. ASM staff will contact you if anything needs to be revised before copyediting and production can begin. Otherwise, you will be notified when your proofs are ready to be viewed.

Data Availability: ASM policy requires that data be available to the public upon online posting of the article, so please verify all links to sequence records, if present, and make sure that each number retrieves the full record of the data. Please separate out the accession numbers for deposited data into a "Data Availability" paragraph instead associated with the relevant methods section. If a new accession number is not linked or a link is broken, provide production staff with the correct URL for the record. If the accession numbers for new data are not publicly accessible before the expected online posting of the article, publication may be delayed; please contact ASM production staff immediately with the expected release date.

Featured Image Submissions: If you would like to submit a potential Featured Image, please email a file and a short legend to mSystems@asmusa.org. Please note that we can only consider images that (i) the authors created or own and (ii) have not been previously published. By submitting, you agree that the image can be used under the same terms as the published article. File requirements: square dimensions (4" x 4"), 300 dpi resolution, RGB colorspace, TIF file format.

Sincerely,
Tricia Van Laar
Editor
mSystems

Reviewer #1 (Comments for the Author):

张博士等人撰写的稿件已根据我所评论的建议进行了成功修改。重要的是，作者对我提出的关于 MbovP0725 蛋白如何调节 P38 和 ERK 蛋白的磷酸化以及 MbovP15 中的 D0725A 氨基酸如何影响磷酸化功能的问题提供了适当的解释。
建议：更正一些小的方法错误和文本编辑。

Reviewer #2 (Comments for the Author):

The authors has response my suggestions point by point.

The manuscript that was written by Dr. Zhang et al has been successfully revised according to the suggestions that I have commented. Importantly, the authors have provided an appropriate explanation for the questions I raised regarding how the MbovP0725 protein regulates the phosphorylation of P38 and ERK proteins, and how the D15A amino acid in MbovP0725 influences the phosphorylation functionality.

Recommendation: corrections to minor methodological errors and text editing.